# CLAMP and Zelda function together to promote *Drosophila* zygotic genome activation

Jingyue Duan[1†‡*], Leila Rieder[2†], Megan M Colonnetta[3], Annie Huang[1], Mary Mckenney[1], Scott Watters[4], Girish Deshpande[1,3], William Jordan[1], Nicolas Fawzi[4], Erica Larschan[1*]

[1]Department of Molecular Biology, Cellular Biology, and Biochemistry, Brown University, Providence, United States; [2]Department of Biology, Emory University, Atlanta, United States; [3]Department of Molecular Biology, Princeton University, Princeton, United States; [4]Department of Molecular Pharmacology, Physiology and Biotechnology, Brown University, Providence, United States

**\*For correspondence:**
jd774@cornell.edu (JD);
erica_larschan@brown.edu (EL)

[†]These authors contributed equally to this work

**Present address:** [‡]Department of Animal Science, College of Agriculture and Life Sciences, Cornell University, Ithaca, United States

**Competing interests:** The authors declare that no competing interests exist.

**Abstract** During the essential and conserved process of zygotic genome activation (ZGA), chromatin accessibility must increase to promote transcription. *Drosophila* is a well-established model for defining mechanisms that drive ZGA. Zelda (ZLD) is a key pioneer transcription factor (TF) that promotes ZGA in the *Drosophila* embryo. However, many genomic loci that contain GA-rich motifs become accessible during ZGA independent of ZLD. Therefore, we hypothesized that other early TFs that function with ZLD have not yet been identified, especially those that are capable of binding to GA-rich motifs such as chromatin-linked adaptor for male-specific lethal (MSL) proteins (CLAMP). Here, we demonstrate that *Drosophila* embryonic development requires maternal CLAMP to (1) activate zygotic transcription; (2) increase chromatin accessibility at promoters of specific genes that often encode other essential TFs; and (3) enhance chromatin accessibility and facilitate ZLD occupancy at a subset of key embryonic promoters. Thus, CLAMP functions as a pioneer factor that plays a targeted yet essential role in ZGA.

## Introduction

During zygotic genome activation (ZGA), dramatic reprogramming occurs in the zygotic nucleus to initiate global transcription and prepare the embryo for further development (*Jukam et al., 2017*). Chromatin changes that activate the zygotic genome during ZGA rely on cooperation among transcription factors (TFs) (*Lee et al., 2014*). However, only pioneer TFs (*Cirillo and Zaret, 1999*; *Mayran and Drouin, 2018*) can bind to closed chromatin before ZGA because most TFs cannot bind to nucleosomal DNA (*Soufi et al., 2015*).

In *Drosophila*, the pioneer TF Zelda (ZLD; zinc-finger early *Drosophila* activator) plays a key role during ZGA (*Liang et al., 2008*). ZLD exhibits several critical characteristics of pioneer TFs, including (1) binding to nucleosomal DNA (*Sun et al., 2015*; *McDaniel et al., 2019*); (2) regulating transcription of early zygotic genes (*Harrison et al., 2011*); and (3) modulating chromatin accessibility to increase the ability of other non-pioneer TFs to bind to DNA (*Schulz et al., 2015*). However, a large subset of ZLD binding sites (60%) are highly enriched for GA-rich motifs and have constitutively open chromatin even in the absence of ZLD (*Schulz et al., 2015*). Therefore, we and others (*Schulz et al., 2015*) hypothesized that other pioneer TFs which directly bind to GA-rich motifs work together with ZLD to activate the zygotic genome.

GAGA-associated factor (GAF; *Farkas et al., 1994*) and chromatin-linked adaptor for male-specific lethal (MSL) proteins (CLAMP; *Soruco et al., 2013*) are two of few known TFs that can bind to

GA-rich motifs and regulate transcriptional activation in *Drosophila* (*Fuda et al., 2015*; *Kaye et al., 2018*). GAF performs several essential functions in early embryos, including chromatin remodeling (*Shimojima et al., 2003*; *Judd et al., 2021*; *Gaskill et al., 2021*) and RNA Pol II recruitment (*Li et al., 2013*; *Fuda et al., 2015*; *Duarte et al., 2016*), and is required for embryonic nuclear divisions (*Bhat et al., 1996*).

CLAMP is a GA-binding TF essential for early embryonic development (*Rieder et al., 2017*) that binds to promoters and plays several vital roles including opening chromatin on the male X chromosome to recruit the MSL dosage compensation complex (*Urban et al., 2017b*; *Rieder et al., 2019*) and activating coordinated regulation of the histone genes at the histone locus (*Rieder et al., 2017*). Therefore, we hypothesized that CLAMP functions with ZLD as a pioneer factor to promote ZGA.

Here, we first demonstrate that depleting maternal CLAMP disrupts transcription of critical early zygotic genes causing significant phenotypic changes in early embryos. Next, we define several mechanisms by which CLAMP regulates ZGA: (1) CLAMP activates zygotic transcription via direct binding to target genes; (2) CLAMP binds directly to nucleosomal DNA and increases chromatin accessibility of promoters of a subset of genes that often encode other essential TFs; and (3) CLAMP and ZLD regulate each other's occupancy at promoters which further regulates the transcription of their target genes. Overall, we determine that CLAMP is an essential pioneer factor that functions with ZLD to regulate ZGA.

## Results

### Depletion of maternal CLAMP disrupts expression of genes that regulate zygotic patterning and cytoskeletal organization in blastoderm embryos

We previously reported that nearly 100% (99.87%) of maternal *clamp* RNAi embryos never hatch and die at early embryonic stages (*Rieder et al., 2017*), demonstrating that maternally deposited CLAMP is critical for embryonic development. To assess embryonic phenotypic patterning after maternal *clamp* depletion, we first identified three key early zygotic genes (*even-skipped*, *runt,* and *neurotactin, Figure 1—figure supplement 1A*) that have significantly (adjusted p<0.05, DESeq2, RNA-seq) reduced expression in early embryos when maternal CLAMP is depleted (*Rieder et al., 2017*). We then used single-molecule fluorescencein situ hybridization (smFISH) for *even-skipped* and *runt,* and immunostaining for Neurotactin (NRT) to determine how the depletion of maternal *clamp* or *zld* alters phenotypic patterning and cytoskeletal integrity in blastoderm stage embryos (*Figure 1*). We validated the knockdown of *clamp* and *zld* in early embryos by qRT-PCR and Western blotting (*Figure 1—figure supplement 1B,C* and *Figure 1—source data 1*).

Both *even-skipped (eve)* and *runt (run)* play an important role in embryonic segmentation (*Manoukian and Krause, 1992*). *Eve* also establishes sharp boundaries between parasegments (*Fujioka et al., 1995*). Strikingly, when maternally deposited *clamp* is depleted, we observed the complete disruption of classic seven stripe pair-rule gene expression patterns using smFISH (*Figure 1A*, middle). Additionally, the nuclei in the embryonic syncytium were disassociated compared to control *egfp* RNAi embryos (*Figure 1A*, left). Furthermore, the expression of *eve* and *run* was significantly reduced in *clamp* maternal depletion embryos that also failed to form sharp stripe boundaries. We observed similar, but slightly stronger, phenotypic changes in *zld* maternal depletion embryos (*Figure 1A*, right), indicating that CLAMP and ZLD have critical roles in establishing embryonic patterning in pre-cellular blastoderm embryos. Moreover, all of the embryos depleted for maternal *clamp* or maternal *zld* (n=10) show defective *eve/run* localization (p<0.05, *Fisher's* exact test).

Next, we used immunostaining to examine the localization of NRT, a cell adhesion glycoprotein that is expressed early during *Drosophila* embryonic cellularization in a lattice surrounding syncytial blastoderm nuclei (*Hortsch et al., 1990*). In *clamp* maternal depletion embryos (*Figure 1B*, middle), we observed dramatically disrupted cellularization and reduced NRT levels. These embryos fail to form the wild-type pattern of cytoskeletal elements, which can be seen in the *egfp* RNAi control embryos (*Figure 1B*, left). Embryos depleted for maternal *zld* also reveal similar patterns of discordant nuclei. More than 50% of embryos depleted for maternal *clamp* (n=23) and 100% of embryos

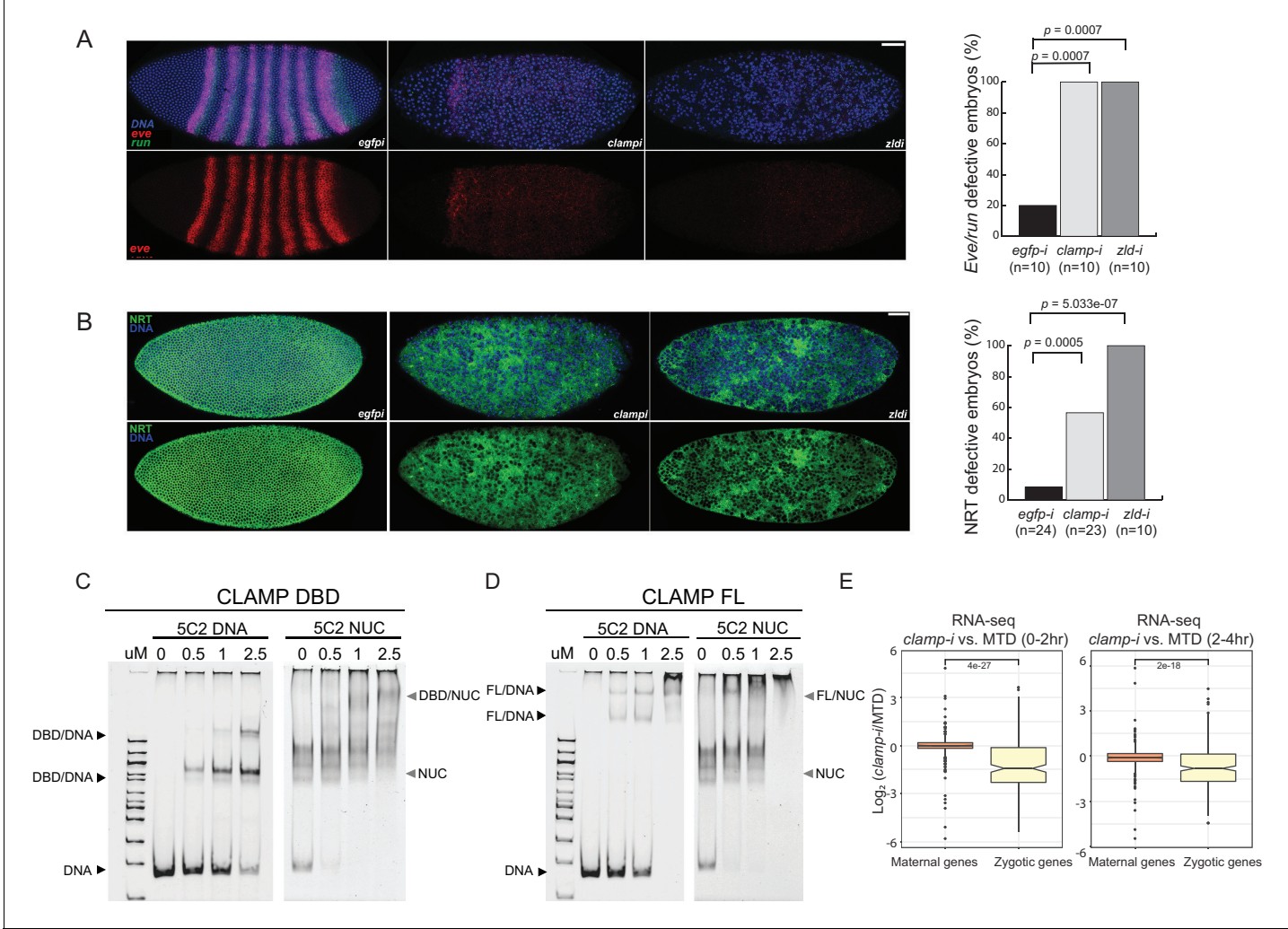

**Figure 1.** Novel pioneer factor CLAMP is essential for early embryonic development. (**A**) Control maternal *egfp* depletion (left), maternal *clamp* depletion (middle), and maternal *zld* depletion (right) syncytial blastoderm stage embryos probed using smFISH for the pair-rule patterning genes *run* (green) and *eve* (red). Embryos were co-labeled with Hoechst (blue) to visualize nuclei. Scale bar represents 10 μm. Quantification (%) of *eve/run* defective embryos is on the right, p-values were calculated with the *Fisher's* exact test; number of embryos is in parentheses. (**B**) Control maternal *egfp* depletion (left), maternal *clamp* depletion (middle), and maternal *zld* depletion (right) syncytial blastoderm stage embryos were assessed for integrity of the developing cytoskeleton using anti-NRT antibody (green) and Hoechst (blue) to label nuclei. Scale bar represents 10 μm. Quantification (%) of NRT defective embryos is on the right, p-values were calculated with the *Fisher's* exact test; number of embryos is in parentheses. (**C**) Electrophoretic mobility shift assay (EMSA) showing the binding of increasing amounts of CLAMP DNA-binding domain (DBD) fused to MBP to 5C2 naked DNA or 5C2 in vitro reconstituted nucleosomes (Nucs). Concentrations (μM) of CLAMP DBD increase from left to right. (**D**) EMSA showing the binding of increasing amounts of full-length (FL) CLAMP (fused to MBP) to 5C2 DNA or 5C2 Nucs. Concentrations (μM) of CLAMP FL increase from left to right. (**E**) Effect of maternal *clamp* RNAi on maternally deposited (orange) or zygotically transcribed (yellow) gene expression log2 (*clamp*-i/MTD) in 0–2 hr (left) or 2–4 hr (right). Maternal versus zygotic gene categories were as defined in *Lott et al., 2011*. p-values of significant expression changes between maternal and zygotic genes were calculated by Mann-Whitney U-test and noted on the plot. CLAMP, chromatin-linked adaptor for male-specific lethal (MSL) proteins; smFISH, single-molecule fluorescence in situ hybridization.

The online version of this article includes the following source data and figure supplement(s) for figure 1:

**Source data 1.** Original western blots and EMSA images.

**Figure supplement 1.** Novel pioneer factor CLAMP is essential for early embryonic development.

depleted for maternal *zld* (n=10) show NRT disruption (*Figure 1B*, right) (p<0.05, *Fisher's* exact test). Overall, smFISH and immunostaining results suggest that both maternally deposited CLAMP and ZLD are essential for early embryonic patterning and development.

## CLAMP binds to nucleosomal DNA in vivo and in vitro

One of the intrinsic characteristics of pioneer TFs is their capacity to bind nucleosomal DNA and compacted chromatin (*Cirillo and Zaret, 1999*). To test the hypothesis that CLAMP is a pioneer TF, we performed electrophoretic mobility shift assays (EMSAs) that test the intrinsic capability of CLAMP to directly interact with nucleosomes in vitro. First, we identified a 240-bp region of the X-linked 5C2 locus (*Figure 1—figure supplement 1D*) that CLAMP binds to in cultured S2 cells and exhibited decreased chromatin accessibility in the absence of CLAMP (*Urban et al., 2017b*). This region is also occupied by a nucleosome (*Figure 1—figure supplement 1D*), suggesting that CLAMP promotes accessibility of this region while binding to nucleosomes.

We then performed in vitro nucleosome assembly using 240 bp of DNA from the 5C2 locus that contains three CLAMP-binding motifs, and we used 5C2 naked DNA as a control. We found that both the CLAMP DNA-binding domain (DBD; *Figure 1C* and *Figure 1—source data 1*) and full-length (FL) protein (*Figure 1D* and *Figure 1—source data 1*) can bind and shift both 5C2 naked DNA and nucleosomes assembled with 5C2 DNA. Increased protein concentration results in a secondary 'super' shift species (*Figure 1C and D* and *Figure 1—source data 1*), indicating that multiple CLAMP molecules may occupy the three CLAMP-binding motifs. Both FL CLAMP and CLAMP DBD are fused to the maltose-binding protein (MBP), which we previously demonstrated does not bind to DNA independent of CLAMP or alter the specificity of CLAMP binding (*Kaye et al., 2018*). Previously, we determined that CLAMP binds specifically to GAGA-repeats in vivo and in vitro (*Soruco et al., 2013*; *Kaye et al., 2018*). Here, we further demonstrate that the zinc-finger protein CLAMP can directly bind to nucleosomal DNA and generates multiple shift species consistent with the potential to bind to multiple binding sites simultaneously.

## CLAMP regulates zygotic genome activation

To define how CLAMP regulates early embryonic patterning, we examined the effect of maternal CLAMP depletion on the expression of maternally deposited or zygotically transcribed genes (*Lott et al., 2011*) using RNA-seq data (*Rieder et al., 2017*). We found that the expression levels of zygotically transcribed genes but not maternally deposited genes were significantly downregulated in embryos lacking CLAMP (p<0.001, Mann-Whitney U-test) (*Figure 1E*). Therefore, CLAMP has a specific effect on the transcription of zygotic genes similar to that which has been previously reported for ZLD (*Liang et al., 2008*; *Harrison et al., 2011*; *McDaniel et al., 2019*) and confirmed in this study (*Figure 1—figure supplement 1E*) using stage 5 embryos lacking maternal ZLD (GSE65837, *Schulz et al., 2015*).

## CLAMP regulates chromatin accessibility in early embryos

An essential characteristic of pioneer TFs is that they can establish and maintain the accessibility of their DNA target sites, allowing other TFs to bind to DNA and activate transcription (*Zaret and Carroll, 2011*; *Iwafuchi-Doi et al., 2016*). We previously used MNase-seq (*Urban et al., 2017b*) to determine that CLAMP guides MSL complex to GA-rich sequences by promoting an accessible chromatin environment on the male X chromosome in cultured S2 cells. Furthermore, GA-rich motifs are enriched in regions that remain accessible in the absence of the pioneer factor ZLD (*Schulz et al., 2015*). Therefore, we hypothesized that CLAMP regulates chromatin accessibility at some ZLD-independent GA-rich loci during ZGA.

To test our hypothesis, we performed the assay for transposase-accessible chromatin using sequencing (ATAC-seq) on 0–2 hr (pre-ZGA) and 2–4 hr (post-ZGA) embryos with wild-type (wt) levels of CLAMP (maternal triple GAL4) driver (MTD) alone (*Ni et al., 2011*) and embryos depleted for maternal CLAMP using RNAi driven by the MTD driver (*clamp-i*). We identified differentially accessible (DA) regions (*Figure 2A* and *Figure 2—source data 1*) by comparing ATAC-seq reads between MTD and *clamp-i* embryos using DiffBind (*Stark and Brown, 2019*). Principal component analysis plots (*Figure 2—figure supplement 1A & D*) show that the first principal component (PC) explains 86–87% of the variation between MTD and *clamp-i* embryos. However, we also observed that 5–6% of the variation among sample replicates is explained by PC2, suggesting the presence of some developmental diversity within sample groups.

Despite some variation among replicates, the high Pearson correlation for DA regions between replicates indicates robust reproducibility of these sites (*Figure 2—figure supplement 1B & E*). We

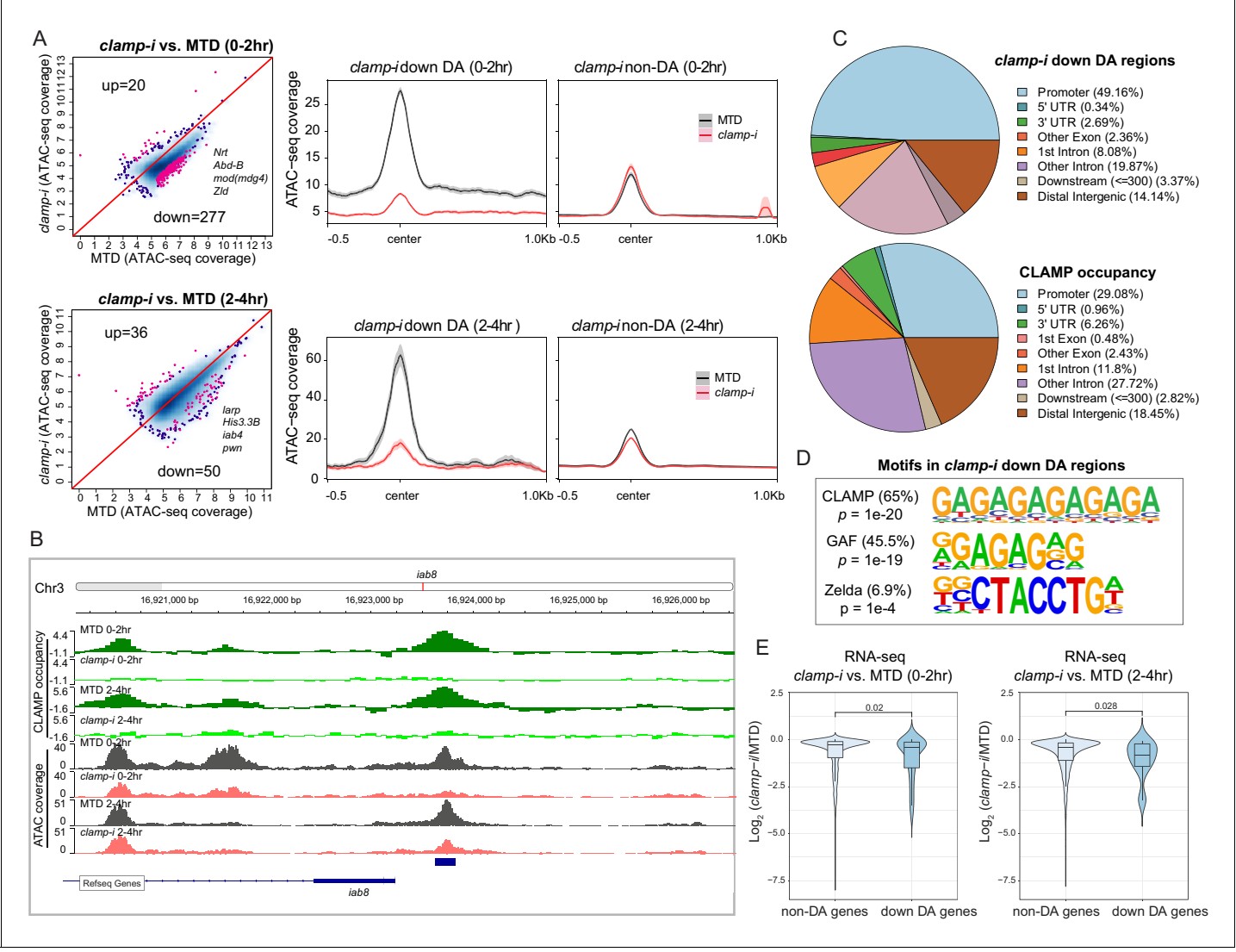

**Figure 2.** CLAMP regulates chromatin accessibility of a subset of the early zygotic genome. (**A**) Differential accessibility (DA) analysis (left) of ATAC-seq from MTD embryos versus *clamp-i* embryos in 0–2 hr or 2–4 hr. Blue dots indicate non-DA sites. Pink dots indicate significant (FDR<0.1) differential peaks after maternal *clamp* RNAi, identified by DiffBind (DESeq2) (DA peaks). The number of peaks and representative genes in each class is noted on the plot. Average ATAC-seq signal (right) in reads per genome coverage (RPGC) 1× normalization in 0–2 hr or 2–4 hr embryos after maternal *clamp* RNAi centered on open chromatin (≤100 bp) peaks identified significant changes upon maternal *clamp* RNAi. (**B**) Example of IGV views of genomic locus *iab-8* bound by CLAMP (ChIP-seq) which shows significantly decreased CLAMP binding and ATAC-seq signal after *clamp* RNAi. (**C**) Genomic features of regions that require CLAMP for chromatin accessibility (*clamp-i* down DA regions, ATAC-seq) compared with all CLAMP binding sites (ChIP-seq occupancy). (**D**) Top motifs enriched in regions that require CLAMP for chromatin accessibility (*clamp-i* down DA regions, ATAC-seq). Enrichment p-value and percentage of sequences are noted. (**E**) Violin plot comparing gene expression (RNA-seq data) in CLAMP-mediated changes and unchanged differential accessibility regions in 0–2 hr or 2–4 hr embryos after maternal *clamp* RNAi. p-values of significant expression changes of CLAMP down-DA and non-DA were calculated by Mann-Whitney U-test and noted on the plot. ChIP-seq, chromatin immunoprecipitation-sequencing; CLAMP, chromatin-linked adaptor for male-specific lethal (MSL) proteins; FDR, false discovery rate.

The online version of this article includes the following source data and figure supplement(s) for figure 2:

**Source data 1.** ATAC-seq read counts in peak region in replicates of MTD and RNAi samples (DiffBind analysis).

**Figure supplement 1.** CLAMP regulates chromatin accessibility throughout ZGA.

identified a subset of genomic regions that exhibit significantly reduced chromatin accessibility in the absence of CLAMP (*Figures 2A* and 0–2 hr: 277; 2–4 hr: 50 and *Figure 2—source data 1*), indicating that chromatin accessibility of these genomic loci (DA sites) requires CLAMP. Moreover, DA sites include promoters of many genes essential for early embryogenesis such as *Nrt, Abd-B, mod*

(*mdg4),* and *zld,* which encodes the ZLD TF (*Figure 2A*). A smaller number of loci (0–2 hr: 20; 2–4 hr: 36) increased their accessibility in the absence of CLAMP (*Figure 2A*). Gene Ontology (GO) analysis (*Figure 2—figure supplement 1C & F*) indicates that CLAMP increases accessibility of chromatin regions that are mainly within DNA-binding, RNA Pol II binding, and enhancer-binding TF encoding genes (*Figure 2—figure supplement 1C & F*). While CLAMP strongly regulates chromatin accessibility non-redundantly with other factors at a subset of genomic loci, CLAMP target genes are key for early development, consistent with the dramatic patterning defects observed after depleting maternal CLAMP (*Figure 1A*).

Furthermore, a subset (26.7% [74/277] at 0–2 hr; 90% [45/50] at 2–4 hr) of DA regions are directly bound by CLAMP, suggesting that CLAMP directly regulates their chromatin accessibility. For example, the *iab8* promoter, which is located within the essential *Drosophila Hox* cluster that controls body plan patterning, is directly bound by CLAMP and shows a reduction in chromatin accessibility after *clamp* RNAi (*Figure 2B*). We also defined the distribution of DA sites and CLAMP binding sites throughout the genome (*Figure 2C*). While DA sites were significantly (p<0.05, *Fisher's* exact test) enriched at promoter regions (49.16%), CLAMP binds almost equally frequently to both promoters (29.08%) and introns that are not first introns (27.72%). Therefore, CLAMP is required to establish or maintain open chromatin largely at promoters, but may also play other roles in intronic regions. Furthermore, motif analysis identified both GA-rich motifs and ZLD motifs enriched at regions that require CLAMP for their accessibility in 0–2 hr embryos (*Figure 2D*). These data suggest that CLAMP may also regulate the accessibility of some ZLD binding sites, a hypothesis that we discuss further below.

We next determined whether CLAMP-mediated chromatin accessibility could specifically drive early transcription by examining the relationship between the chromatin accessibility (DA, ATAC-seq) changes and gene expression of the nearest gene as measured by RNA-seq (*Rieder et al., 2017*). We observed significant (p<0.05, Mann-Whitney U-test) reduction in expression after maternal CLAMP depletion of genes at which CLAMP mediates chromatin accessibility (DA genes) compared with genes at which CLAMP does not regulate chromatin accessibility (non-DA genes) (*Figure 2E*). Overall, our results indicate that CLAMP promotes chromatin accessibility and transcription of a subset of other essential TF genes during ZGA, which is consistent with the extensive developmental defects caused by maternal CLAMP depletion.

## CLAMP and ZLD regulate each other's binding to a subset of promoters

To directly determine how CLAMP and ZLD impact each other's binding, we performed ChIP-seq for CLAMP and ZLD in control MTD embryos and embryos that were maternally depleted for each factor with RNAi at the same two time points we used for our ATAC-seq experiments: before ZGA (0–2 hr) and during and after ZGA (2–4 hr) (*Figure 3A–B*, *Figure 3—figure supplement 1A–B* and *Table 1*). Overall, there are more ZLD peaks (0–2 hr: 6974; 2–4 hr: 8035) across the whole genome than CLAMP peaks (0–2 hr: 4962, 2–4 hr: 7564) in control MTD embryos. As we hypothesized, CLAMP and ZLD peaks significantly overlap (p<0.05, hypergeometric test, N=15,682 total fly genes) (*Figure 3—figure supplement 1C*).

Next, we defined the sites that showed differential binding (DB; *Figure 3C–D* and *Table 1*) of CLAMP and ZLD in the absence of each other's maternally deposited mRNA using DiffBind (*Stark and Brown, 2019*). We found a significant reduction of ZLD binding in the absence of CLAMP: there were 274 (0–2 hr) and 1289 (2–4 hr) sites where ZLD binding decreased in *clamp-i* embryos compared to MTD controls (down-DB) (*Figure 3C*, *Figure 3—figure supplement 1D*, and *Table 1*). Fewer ZLD binding sites increased in occupancy after *clamp* RNAi: 8 (0–2 hr) and 233 (2–4 hr) sites (up-DB). 390 (0–2 hr) and 30 (2–4 hr) CLAMP down-DB sites were found upon loss of ZLD (*Figure 3D*, *Figure 3—figure supplement 1E*, and *Table 1*). We identified very few sites where CLAMP occupancy increases after zld RNAi (up-DB sites: 0–2 hr: 54, 2–4 hr: 3). Moreover, depletion of either maternal zld or clamp mRNA altered the genomic distribution of CLAMP and ZLD: the most common pattern we observed was that promoter-bound peaks were lost (down-DB) and peaks in introns were gained (up-DB) (*Figure 3—figure supplement 3*).

The CLAMP down-DB sites and the ZLD down-DB sites also significantly overlap with each other (p<0.05, hypergeometric test, N=15,682 total fly genes) at both time points (*Figure 3—figure supplement 1F*). For example, *iab-8*, an essential *Hox* cluster gene at which CLAMP regulates chromatin

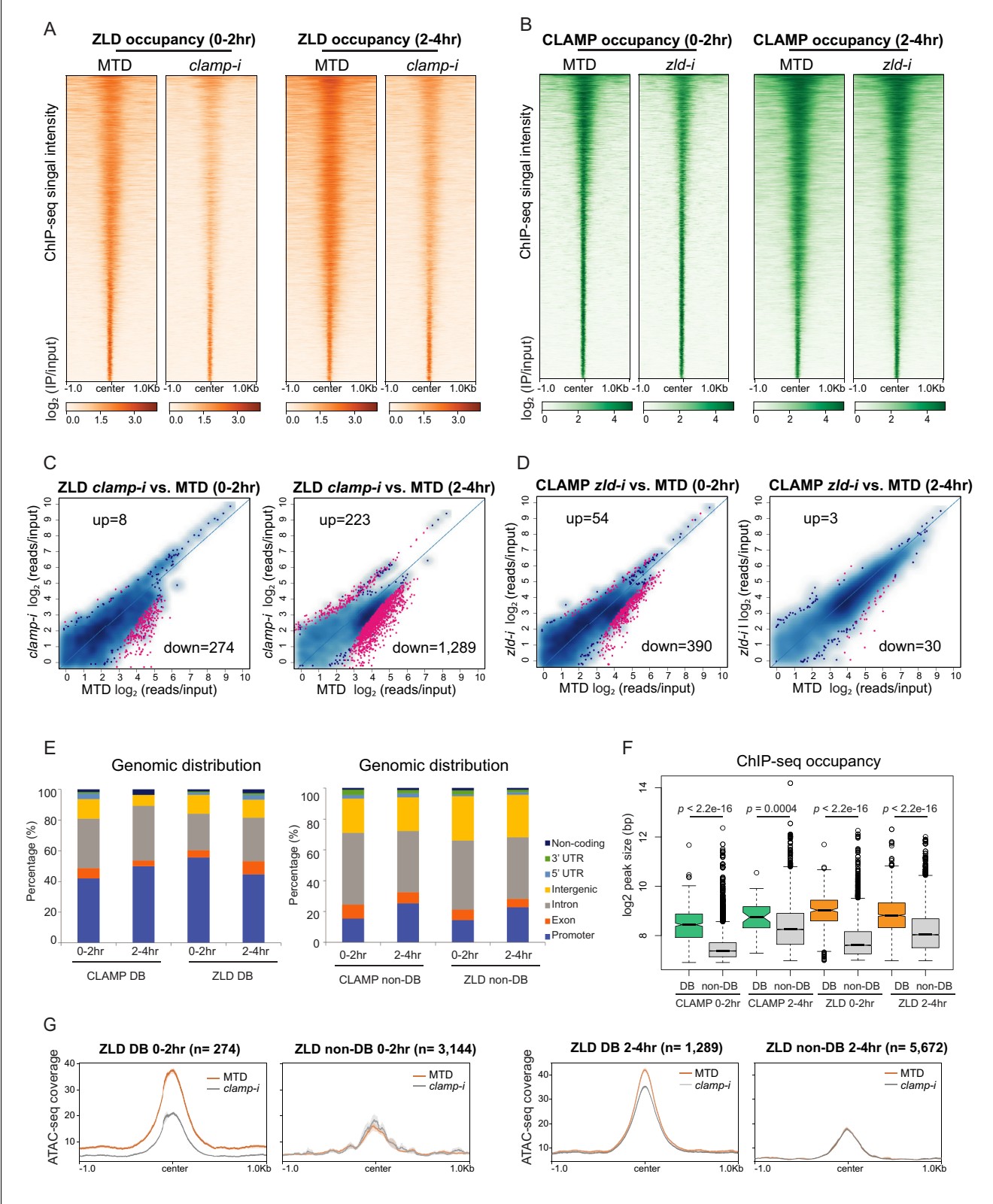

**Figure 3.** CLAMP and ZLD depend on each other for binding at a subset of sites. (**A**) ZLD occupancy in 0–2 hr and 2–4 hr MTD and maternal *clamp* RNAi embryos. Data is displayed as a heatmap of z-score normalized ChIP-seq (log$_2$ IP/input) reads in a 2-kb region centered at each peak. Peaks in each class are arranged in order of decreasing z-scores in control MTD embryos. (**B**) CLAMP occupancy in 0–2 hr and 2–4 hr MTD and maternal *zld* RNAi embryos. Data is displayed as a heatmap of z-score normalized ChIP-seq (log$_2$ IP/input) reads in a 2-kb region centered around each peak. Peaks

*Figure 3 continued on next page*

*Figure 3 continued*

in each class are arranged in order of decreasing z-scores in control MTD embryos. (**C**) Differential binding (DB) analysis of ZLD ChIP-seq. Mean difference (MA) plots of ZLD peaks in MTD embryos versus *clamp*-i embryos in 0–2 hr (left) or 2–4 hr (right). Blue dots indicate non-DB sites. Pink dots indicate significant (FDR<0.05) differential peaks identified by DiffBind (DESeq2). The number of peaks changed in each direction is noted on the plot. (**D**) DB analysis of CLAMP ChIP-seq. MA plots of CLAMP peaks from MTD embryos versus *zld*-i embryos in 0–2 hr (left) or 2–4 hr (right). Blue dots indicate non-DB sites. Pink dots indicate significant (FDR<0.05) DB peaks identified by DiffBind (DESeq2). Number of peaks in each direction is noted on the plot. (**E**) Stacked bar plots of CLAMP and ZLD down-DB (left) and CLAMP and ZLD non-DB peaks (right) distribution fraction in the *Drosophila* genome (dm6) in 0–2 hr and 2–4 hr embryos. (**F**) Box plot of the peak sizes in CLAMP and ZLD down-DB and non-DB peaks in 0–2 hr and 2–4 hr embryos. p-values of significant size difference between down-DB and non-DB peaks were calculated by Mann-Whitney U-test and noted on the plot. (**G**) Average profiles of ATAC-seq signal coverage show chromatin accessibility at ZLD down-DB (orange line) and non-DB (gray line) sites in 0–2 hr (left panel) or 2–4 hr (right panel) MTD and *clamp*-i embryos. Number of sites is noted on the plot. ChIP-seq, chromatin immunoprecipitation-sequencing; CLAMP, chromatin-linked adaptor for male-specific lethal (MSL) proteins.

The online version of this article includes the following figure supplement(s) for figure 3:

**Figure supplement 1.** CLAMP and ZLD depend on each other for chromatin binding.
**Figure supplement 2.** CLAMP and ZLD depend on each other for chromatin binding.
**Figure supplement 3.** CLAMP and ZLD depend on each other for chromatin binding.

accessibility (DA), is also one of the 95 genomic loci at which CLAMP and ZLD promote each other's occupancy (*Figure 3—figure supplement 1F–G*).

To further understand how CLAMP and ZLD bind to dependent (down-DB) and independent (non-DB) sites, we determined the genomic distribution and size of occupancy of these two types of sites. Overall, dependent peaks (down-DB) are much broader in size and are located at promoters (*Figure 3E*, *Figure 3—figure supplement 1H* and *Figure 3—figure supplement 2A*). In contrast, independent sites (non-DB) are narrower and located within introns (*Figure 3E*, *Figure 3—figure supplement 1H* and *Figure 3—figure supplement 2A*). On average, the peak size of dependent sites (down-DB: 400–500 bp) is almost double that of independent sites (non-DB: 200–250 bp) with significant differences in peak size for both TFs at both time points (p<0.001, Mann-Whitney U-test) (*Figure 3F*).

Previous proteomic studies (*Urban et al., 2017a*; *Hamm et al., 2017*; *Hamm et al., 2015*) found no evidence that CLAMP and ZLD directly contact each other at the protein level, suggesting that CLAMP and ZLD regulate each other via binding to their DNA motifs. Therefore, we analyzed the motifs enriched at dependent (down-DB) and independent (non-DB) sites. We found that dependent sites are enriched for motifs specific for the required protein, which are not present at the independent sites (*Figure 3—figure supplement 2B*). For example, the ZLD motif is only enriched at sites where CLAMP requires ZLD for binding (CLAMP down-DB) but not at sites where CLAMP binds independently of ZLD (CLAMP non-DB). Similarly, the CLAMP motif is only enriched at sites where ZLD requires CLAMP for binding (ZLD down-DB) (*Figure 3—figure supplement 2B*). Therefore, the

**Table 1.** The number of total and differentially bound peaks for CLAMP and ZLD in control MTD, *clamp-i*, and *zld-i* embryos.

| ChIP-seq peaks | CLAMP | | | ZLD | | |
|---|---|---|---|---|---|---|
| | **MTD** | ***clamp-i*** | ***zld-i*** | **MTD** | ***clamp-i*** | ***zld-i*** |
| 0–2 hr | 4962 | 3488 | 4746 | 6974 | 3687 | 4650 |
| 2–4 hr | 7564 | 4064 | 8279 | 8035 | 4687 | 6420 |
| Differential binding (DiffBind, DEseq2) | MTD versus *zld-i* | | | MTD versus *clamp-i* | | |
| | Up-DB | Down-DB | Non-DB | Up-DB | Down-DB | Non-DB |
| 0–2 hr | 54 | 390 | 4184 | 8 | 274 | 3144 |
| 2–4 hr | 3 | 30 | 7351 | 223 | 1289 | 5672 |

The online version of this article includes the following source data for Table 1:

**Source data 1.** ChIP-seq read counts in peak regions in replicates of MTD and RNAi samples (DiffBind analysis).Page 1. ZLD ChIP-seq in *clamp-i* versus MTD in 0–2 hr embryos. Page 2. ZLD ChIP-seq in *clamp-i* versus MTD in 2–4 hr embryos. Page 3. CLAMP ChIP-seq in *zld*-i versus MTD in 0–2 hr embryos. Page 4. CLAMP ChIP-seq in *zld-i* versus MTD in 2–4 hr embryos.

presence of specific CLAMP and ZLD motifs correlates with their ability to promote each other's binding.

Given the cooperative relationship between CLAMP and ZLD binding to chromatin, we measured chromatin accessibility (ATAC-seq coverage) changes at their dependent and independent sites that we defined from ChIP-seq data (*Figure 3G* and *Figure 3—figure supplement 2C*). We found the average ATAC-seq signals were significantly reduced at sites where ZLD is dependent on CLAMP to bind (ZLD down-DB sites) in *clamp-i* embryos compared to MTD controls (*Figure 3G*). Furthermore, the accessibility at sites where ZLD binds independently of CLAMP (ZLD non-DB) is lower than that at ZLD DB sites but remains unchanged upon *clamp* RNA-i (*Figure 3G*). Therefore, the chromatin accessibility changes we observe over broader regions are enriched at specific loci where CLAMP promotes ZLD binding.

Sites where ZLD regulates CLAMP binding (CLAMP down-DB) have high chromatin accessibility while sites where CLAMP binds independently (CLAMP non-DB) of ZLD showed low chromatin accessibility (*Figure 3—figure supplement 2C*). Interestingly, accessibility slightly increases upon the loss of ZLD at sites where CLAMP requires ZLD for binding at 0–2 hr (*Figure 3—figure supplement 2C*). However, an active TF binding to DNA can prevent Tn5 cleavage at genomic regions (*Yan et al., 2020*). Therefore, loss of ZLD and CLAMP binding could result in a perceived accessibility increase, as measured by ATAC-seq, which does not necessarily reflect a repressive function for ZLD.

In summary, CLAMP and ZLD increase each other's occupancy by binding to their motifs and altering chromatin accessibility. These data support a model in which CLAMP and ZLD increase each other's occupancy at promoters of a subset of genes that often encode other TFs.

## CLAMP and ZLD function together to regulate transcription during ZGA

CLAMP and ZLD both specifically regulate zygotic transcription (*Figure 1E* and *Figure 1—figure supplement 1E*; *Liang et al., 2008*; *Harrison et al., 2011*; *McDaniel et al., 2019*; *Rieder et al., 2017*). To further understand how CLAMP and ZLD function to regulate ZGA, we compared the transcriptional roles of CLAMP and ZLD in early embryos at genes that have different temporal expression patterns as defined in *Li X-Y et al., 2014*. We found that both CLAMP and ZLD are present at genes expressed throughout ZGA although CLAMP binding is more often present at mid- and late-transcribed zygotic genes (categories defined in *Li X-Y et al., 2014*), while ZLD binding is more often present at early transcribed zygotic genes (*Figure 4A–B*).

We next asked whether the ability of CLAMP to bind to genes directly regulates zygotic gene activation by integrating ChIP-seq with RNA-seq data (*Schulz et al., 2015*; *Rieder et al., 2017*). We found that genes strongly bound or weakly bound by CLAMP (ChIP-seq data) showed a significant (p<0.001, Mann-Whitney U-test) level of gene expression reduction after *clamp* RNAi (*Rieder et al., 2017*) compared to unbound genes (*Figure 4C*). We also observed a significant (p<0.001, Mann-Whitney U-test) change in gene expression in maternal *zld-i* embryos (*Schulz et al., 2015*) for the genes that are strongly bound by ZLD (ChIP-seq data) (*Figure 4D*). Also, the magnitude of the transcriptional changes is similar for genes that are bound by CLAMP or ZLD. Together, these data indicate that CLAMP regulates the transcription of zygotic genes by directly binding to target genes.

To investigate whether CLAMP and ZLD could regulate each other's binding to precisely drive the transcription of target genes, we plotted the gene expression changes caused by depleting maternal *zld* or *clamp* at the genes closest to where they regulate each other's binding (*Figure 4E* and *Figure 4F*). The depletion of maternal zld significantly (*p*=4.3E−5, Mann-Whitney U-test) reduces the expression of genes where ZLD regulates CLAMP binding (down-DB) more than sites where CLAMP binds independently of ZLD (non-DB) (*Figure 4E*). Therefore, ZLD may specifically regulate zygotic genes at which ZLD promotes CLAMP binding. Also, compared to genes where ZLD binds independent of CLAMP, genes where ZLD binding is regulated by CLAMP had a significant (*p*<0.001, Mann-Whitney U-test) expression reduction after clamp RNAi at both 0–2 hr and 2–4 hr time points (*Figure 4F*). Thus, CLAMP may regulate the transcription of genes targeted by ZLD by promoting ZLD binding.

Furthermore, sites where CLAMP and ZLD require each other for binding are enriched for motifs specific for the required protein (*Figure 3—figure supplement 2B*). Therefore, the presence of

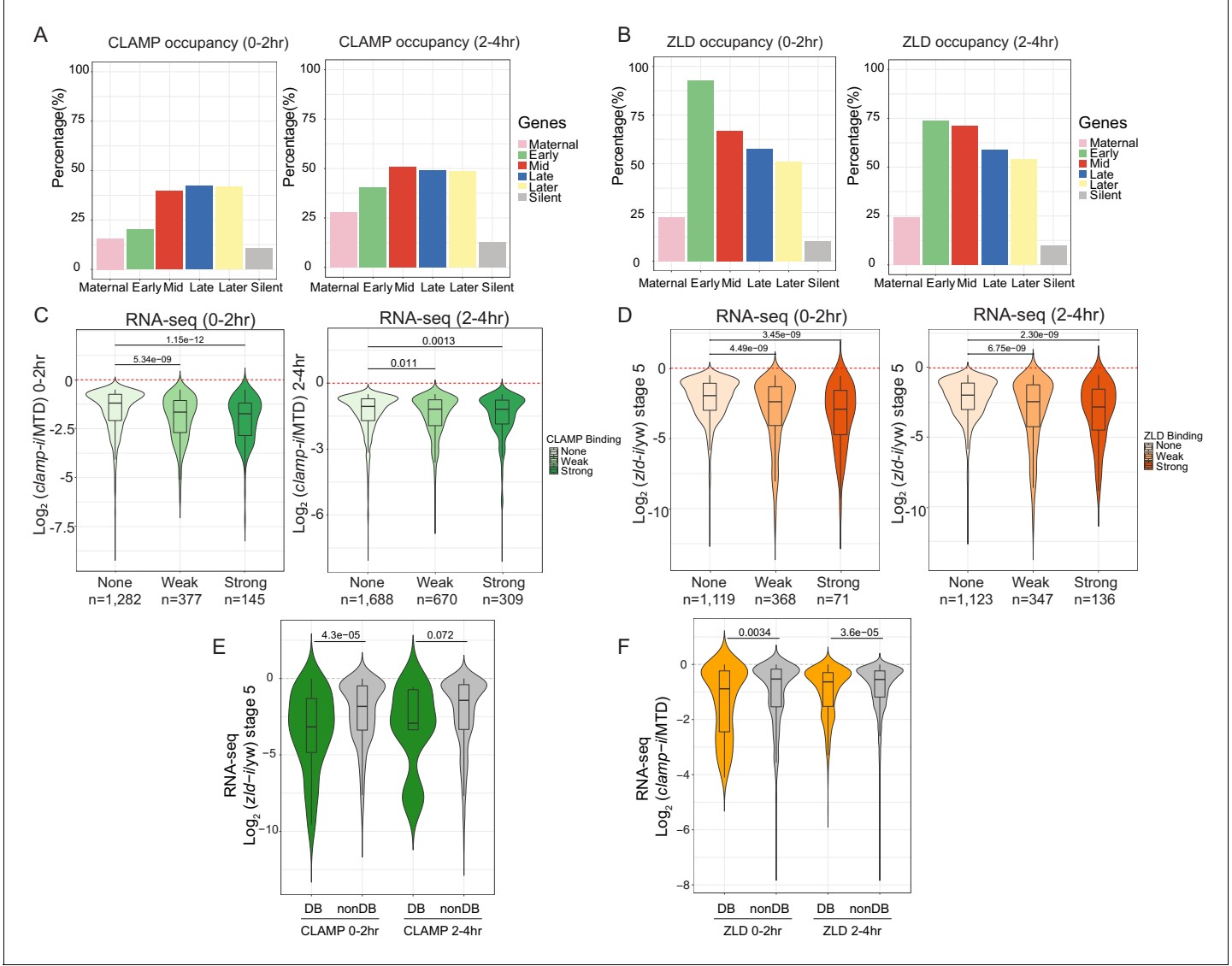

**Figure 4.** CLAMP and ZLD function together in zygotic genome activation. (**A**) Percentage of CLAMP binding sites in 0–2 hr and 2–4 hr embryos distributed in maternal (n=646), early (n=69), mid- (n=73), late- (n=104), later (n=74), and silent (n=921) genes (peaks within a 1-kb promoter region and gene body). Gene categories were defined in *Li X-Y et al., 2014*. (**B**) Percentage of ZLD binding sites in 0–2 hr and 2–4 hr embryos distributed in maternal (n=646), early (n=69), mid- (n=73), late- (n=104), later (n=74), and silent (n=921) genes (peaks within a 1-kb promoter region and gene body). Gene categories were defined in *Li X-Y et al., 2014*. (**C**) Gene expression changes caused by maternal *clamp* RNAi (*Rieder et al., 2017*) at genes with strong, weak, and no CLAMP binding as measured by ChIP-seq in 0–2 hr (left) or 2–4 hr (right) embryos. p-values of significant expression changes of CLAMP bindings were calculated by Mann-Whitney U-test and noted on the plot. (**D**) Gene expression changes caused by maternal *zld* RNAi (*Schulz et al., 2015*) at genes with strong, weak, and no ZLD binding as measured by ChIP-seq in 0–2 hr (left) or 2–4 hr (right) embryos. p-values of significant expression changes of ZLD bindings were calculated by Mann-Whitney U-test and noted on the plot. (**E**) Gene expression changes caused by maternal *zld* RNAi (*Schulz et al., 2015*) at genes with CLAMP down-DB and non-DB that defined in wt versus *zld*- 0–2 hr and 2–4 hr embryos ChIP-seq. p-values of significant expression changes of CLAMP down-DB and non-DB were calculated by Mann-Whitney U-test and noted on the plot. (**F**) Gene expression changes caused by maternal *clamp* RNAi (*Rieder et al., 2017*) at genes with ZLD down-DB and non-DB that defined in MTD versus *clamp*-i 0–2 hr and 2–4 hr embryos ChIP-seq. p-values of significant expression changes of ZLD down-DB and non-DB were calculated by Mann-Whitney U-test and noted on the plot. ChIP-seq, chromatin immunoprecipitation-sequencing; CLAMP, chromatin-linked adaptor for male-specific lethal (MSL) proteins; DB, differential binding; wt, wild-type.

specific CLAMP and ZLD motifs correlates with their ability to promote each other's binding which further regulates the expression of each other's target genes.

## CLAMP and ZLD regulate gene expression via modulating chromatin accessibility

To determine how direct binding of CLAMP and ZLD relates to zygotic chromatin accessibility, we integrated ChIP-seq and ATAC-seq data. First, we defined four classes of CLAMP-related peaks (DA with CLAMP, DA without CLAMP, non-DA with CLAMP, and non-DA without CLAMP in *Table 2* and *Table 2—source data 1*). We also obtained ZLD-related ATAC-seq data (*Hannon et al., 2017*; *Soluri et al., 2020*) that was generated from embryos laid by wt mothers or mothers with *zld* germline clones (*zld-*) at the nuclear cycle 14 (NC14) +12 min time point and integrated it with ChIP-seq data from the closest time point from this study (0–2 hr embryos). In this way, we defined four classes of genomic loci related to ZLD: DA with ZLD, DA without ZLD, non-DA with ZLD, and non-DA without ZLD (*Table 2* and *Table 2—source data 1*).

Next, we generated heatmaps to visualize ATAC-seq read coverage and CLAMP and ZLD ChIP-seq occupancy at their related classes of loci in MTD (wt), *clamp-i*, and *zld-i* (*zld-*) embryos (*Figure 5—figure supplement 1A*). As expected, MTD and *clamp-i* embryo heatmaps revealed that CLAMP-related DA regions (DA with CLAMP, DA without CLAMP) show a significant decrease in accessibility in embryos lacking CLAMP. Regions dependent on ZLD to open (DA with ZLD, DA without ZLD) also show a significant accessibility reduction in the absence of ZLD. Moreover, ChIP-seq read enrichment for protein binding in each RNAi or germline clone embryo class corresponds to our classification.

Interestingly, both CLAMP and ZLD protein occupancy on chromatin were reduced when the other TF was depleted, especially at regions where the bound protein is not required for chromatin accessibility (non-DA) (*Figure 5—figure supplement 1A*). For example, ZLD occupancy was reduced upon *clamp* RNAi at ZLD non-DA regions which are bound by ZLD to a level that resembles the ZLD occupancy in *zld-i* embryos (*Figure 5—figure supplement 1A*). We also found that CLAMP is enriched (ChIP-seq signal) at these ZLD non-DA regions, supporting our hypothesis that CLAMP facilitates ZLD occupancy at some of these loci.

To determine the relationship between CLAMP and ZLD in regulating chromatin accessibility at loci bound by both factors, we identified the subset of genomic loci (n=525) that co-bound both CLAMP and ZLD and have open zygotic chromatin (*Figure 5A*, *Table 2* and *Table 2—source data 1*). We divided these co-bound loci into four types: 1% (n=5) of these loci show reduced accessibility after either *clamp-i* or *zld-i* (Type I, both DA, *Figure 5A*); 23 and 123 loci are specifically dependent on CLAMP or ZLD for their accessibility, respectively (Type II, CLAMP-DA and Type III, ZLD-DA, *Figure 5A*); the majority (374 out of 525) of CLAMP ZLD co-bound loci remain open when either protein is absent (Type IV, both non-DA, *Figure 5A*), suggesting either that CLAMP and ZLD function redundantly at some of these loci or that the presence of other TFs regulate their accessibility.

Notably, at sites where CLAMP is required for chromatin accessibility (Type II, CLAMP-DA, n=23), ZLD occupancy is entirely ablated in *clamp-i* embryos (*Figure 5A*). CLAMP occupancy levels are also reduced after maternal *zld* RNAi at sites where ZLD is required for chromatin accessibility (Type III,

**Table 2.** The number of peaks in four types of CLAMP or ZLD mediated regions.

| ATAC-seq peaks (0–2 hr) | DA w/ CLAMP | DA w/o CLAMP | Non-DA w/ CLAMP | Non-DA w/o CLAMP |
|---|---|---|---|---|
| 16,597 | 74 | 203 | 1239 | 15,081 |
| ATAC-seq peaks (*Hannon et al., 2017*) (NC14 +12 min) | DA w/ ZLD | DA w/o ZLD | Non-DA w/ ZLD | Non-DA w/o ZLD |
| 19,146 | 976 | 2782 | 2010 | 13,378 |
| CLAMP ZLD co-bound open chromatin regions | Type I (Both DA) | Type II (CLAMP DA) | Type III (ZLD DA) | Type IV (Both non-DA) |
| 525 | 5 | 23 | 123 | 374 |

The online version of this article includes the following source data for Table 2:

Source data 1. Peaks locations in each CLAMP or ZLD-related category.Page 1 Type I (n=5): both DA, CLAMP ZLD co-bound Page 2 Type II (n=23): CLAMP DA and ZLD non-DA, CLAMP ZLD co-bound Page 3 Type III (n=88): ZLD DA and CLAMP non-DA, CLAMP ZLD co-bound Page 4 Type IV (n=434): both non-DA, CLAMP ZLD co-bound Page 5 DA with CLAMP 0–2 hr; Page 6 DA without CLAMP 0–2 hr; Page 7 non-DA with CLAMP 0–2 hr; Page 8 non-DA without CLAMP 0–2 hr; Page 9 DA with ZLD NC14 +12 min; Page 10 DA without ZLD NC14 +12 min; Page 11 non-DA with ZLD NC14 +12 min; Page 12 non-DA, without ZLD NC14 +12 min.

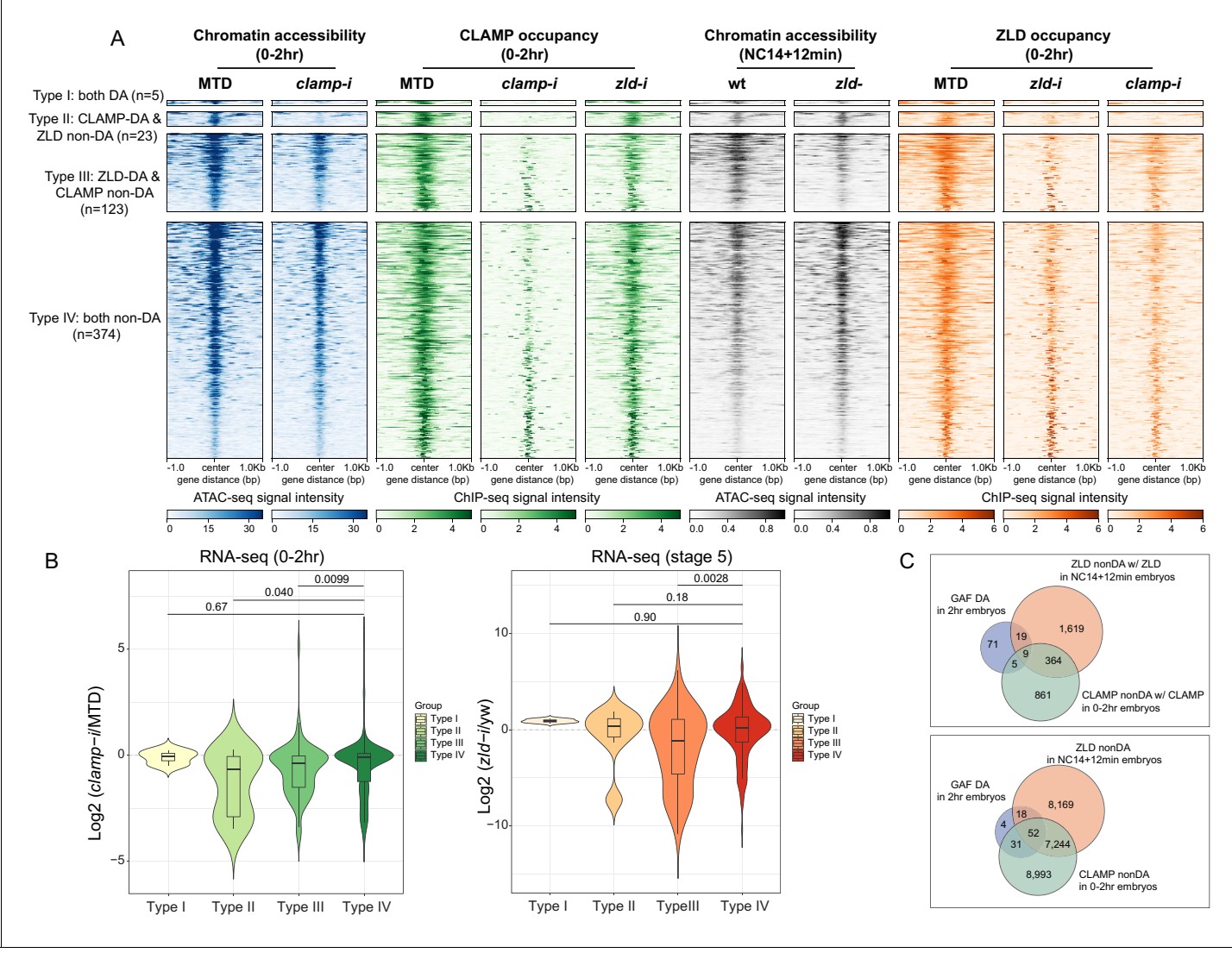

**Figure 5.** CLAMP and ZLD regulate gene expression via modulating chromatin accessibility. (**A**) Four classes of CLAMP and ZLD co-bound peaks defined by combining ATAC-seq (this study or *Hannon et al., 2017*; *Soluri et al., 2020*) and ChIP-seq peaks in 0–2 hr MTD and RNAi embryos. Data is displayed as a heatmap of z-score normalized ATAC-seq and ChIP-seq reads in a 2-kb region centered around each peak. Peaks in each class are arranged in order of decreasing z-scores in control MTD embryos. Type I (n=5): both DA, differentially accessible regions which depend on CLAMP or ZLD; has both proteins bound. Type II (n=23): CLAMP-DA and ZLD non-DA, differentially accessible regions which depend on CLAMP, not on ZLD; has both proteins bound. Type III (n=123): ZLD-DA and CLAMP non-DA, differentially accessible regions which depend on ZLD, not on CLAMP; has both proteins bound. Type IV (n=374): both non-DA, accessibility independent from CLAMP or ZLD; has both proteins bound. (**B**) Left: Gene expression changes caused by maternal *clamp* RNAi (*Rieder et al., 2017*) in 0–2 hr embryos at genes fall into four classes of CLAMP and ZLD co-bound peaks. p-values of significant expression changes among classes were calculated by Mann-Whitney U-test and noted on the plot. Right: Gene expression changes caused by maternal *zld* RNAi (*Schulz et al., 2015*) in stage 5 embryos at genes fall into four classes of CLAMP and ZLD co-bound peaks. p-values of significant expression changes among classes were calculated by Mann-Whitney U-test and noted on the plot. (**C**) Upper panel: Venn diagram showing the number of overlapping sites between GAF-dependent DA sites (*Gaskill et al., 2021*), ZLD non-DA with ZLD bound, and CLAMP non-DA with CLAMP bound peaks. Lower panel: Venn diagram showing the number of overlapping sites between GAF-dependent DA sites (*Gaskill et al., 2021*), ZLD non-DA, and CLAMP non-DA peaks. ChIP-seq, chromatin immunoprecipitation-sequencing; CLAMP, chromatin-linked adaptor for male-specific lethal (MSL) proteins.

The online version of this article includes the following figure supplement(s) for figure 5:

**Figure supplement 1.** CLAMP-mediated chromatin accessibility is correlated with CLAMP and ZLD binding.

ZLD-DA, n=123). Overall, we observed that CLAMP and/or ZLD occupancy is reduced at most of their co-bound regions when either one of the TFs is depleted, which is consistent with their inter-dependent binding relationship. Moreover, *clamp-i* has a stronger impact on ZLD occupancy than *zld-i* has on CLAMP occupancy.

To assess how CLAMP/ZLD-modulated chromatin accessibility impacts transcription, we examined the effect of maternal *clamp* (*Rieder et al., 2017*) or *zld* (*Schulz et al., 2015*) depletion on expression (RNA-seq data) of genes that fall into the four types of CLAMP/ZLD co-occupied sites (*Figure 5B*). We found that the expression levels of genes (Type II, CLAMP-DA, n=23) that require CLAMP for chromatin accessibility are significantly (p<0.05, Mann-Whitney U-test) downregulated in embryos lacking CLAMP compared to the Type IV (both non-DA) CLAMP and ZLD-independent group (n=374) (*Figure 5B*). Genes (Type III, ZLD-DA, n=123) dependent on ZLD for their accessibility also show a significant (p<0.001, Mann-Whitney U-test) reduction in expression upon maternal CLAMP depletion, suggesting CLAMP also might contribute to the regulation of genes at which ZLD regulates chromatin accessibility, likely by increasing ZLD binding.

In embryos depleted for maternal ZLD (*Schulz et al., 2015*), we found genes that fall into the Type III (ZLD-DA) ZLD-mediate chromatin accessibility group significantly (p<0.001, Mann-Whitney U-test) decreased in expression (*Figure 5B*), compared with the Type IV (both non-DA, n=374) group. Interestingly, genes within the CLAMP and ZLD-independent Type IV (both non-DA, n=374) group do not show significant expression fold changes after depleting either maternal *clamp* or *zld*, supporting the hypothesis that CLAMP and ZLD function redundantly at these loci and/or other proteins play a major role in regulating chromatin accessibility and transcription of these genes.

Motif analysis demonstrates that CLAMP and ZLD motifs are enriched at genomic loci that are regulated by each factor as well as independent sites (Type IV), in addition to the motif for another GA-binding protein, GAF (*Figure 5—figure supplement 1B*). We next determined whether GAF alters chromatin accessibility at loci at which depletion of CLAMP or ZLD individually alters accessibility (Type IV) and is bound by all three factors. Indeed, we found that approximately 10% of loci that require GAF for their chromatin accessibility (n=104) (*Gaskill et al., 2021*) overlap with regions where depleting CLAMP or ZLD individually does not alter accessibility (CLAMP non-DA and/or ZLD non-DA) (*Figure 5C*, upper panel). When we do not require occupancy of ZLD and CLAMP at their non-DA sites, the overlap with the GAF-dependent regions is approximately 97% (*Figure 5C*, lower panel). These results suggest GAF might function at these CLAMP/ZLD independent sites, supporting a model in which multiple TFs coordinately regulate early zygotic chromatin accessibility during ZGA (*Hamm and Harrison, 2018*).

Together, our results reveal the CLAMP and ZLD regulate chromatin accessibility, which alters the occupancy of both factors and regulates zygotic transcription. Furthermore, GAF and/or other TFs might function at sites that are not altered by depleting CLAMP or ZLD individually, suggesting that multiple TFs promote chromatin accessibility during ZGA. It is also possible that CLAMP and ZLD are functionally redundant at the subset of genomic loci at which they regulate each other's occupancy, but depleting either factor individually is not sufficient to alter chromatin and expression.

## Discussion

Two questions central to early embryogenesis of all metazoans are how and where do early TFs work together to drive chromatin changes and ZGA. Here, we defined a novel function of CLAMP as a new pioneer TFs that has a targeted yet essential function in early embryonic development. We found that CLAMP directly binds to nucleosomal DNA (*Figure 1*), establishes and/or maintains chromatin accessibility at promoters of genes that often encode other TFs (*Figure 2*), and facilitates the binding of ZLD to promoters (*Figure 3*) to regulate activation of zygotic gene transcription (*Figure 4*). We discovered that CLAMP and ZLD regulate each other's binding via mediating chromatin accessibility which further regulates their target gene expression (*Figure 5*). Overall, we provide new insight into how CLAMP and ZLD function together to enhance each other's occupancy and increase chromatin accessibility, which drives ZGA.

## CLAMP and ZLD act together to define an open chromatin landscape and activate transcription in early embryos

We defined multiple classes of CLAMP-dependent and ZLD-dependent genomic loci in early embryos, which provides insight into how CLAMP and ZLD regulate chromatin accessibility and zygotic transcription during ZGA (*Figure 6*): (1) CLAMP promotes ZLD enrichment at sites where CLAMP increases chromatin accessibility and further regulates ZLD target gene expression. These loci remain open and transcriptionally active even upon ZLD depletion. (2) ZLD facilitates CLAMP occupancy at sites where ZLD regulates chromatin accessibility and promotes CLAMP target gene expression. When maternal CLAMP is depleted, these loci remain accessible and genes are actively transcribed. (3) GAF and/or other TFs could play major roles in opening chromatin at locations co-bound by CLAMP and ZLD but that are not altered in accessibility after depleting CLAMP or ZLD individually. CLAMP and ZLD could also function redundantly at some of these loci because they alter each other's occupancy at these loci but do not change accessibility or expression after depletion of either maternal CLAMP or ZLD individually. Overall, our data suggest that CLAMP functions with ZLD regulate chromatin accessibility and gene expression of the early zygotic genome.

Although we have demonstrated an instrumental role for CLAMP in defining a subset of the open chromatin landscape in early embryos, our data show that CLAMP does not increase chromatin accessibility at promoters of all zygotic genes independent of ZLD. Consistent with our results in the early embryo, CLAMP regulates chromatin accessibility at only a few hundred genomic loci in male S2 (258 sites) and female Kc (102 sites) cell lines. Unlike ZLD, which plays a global role in regulating

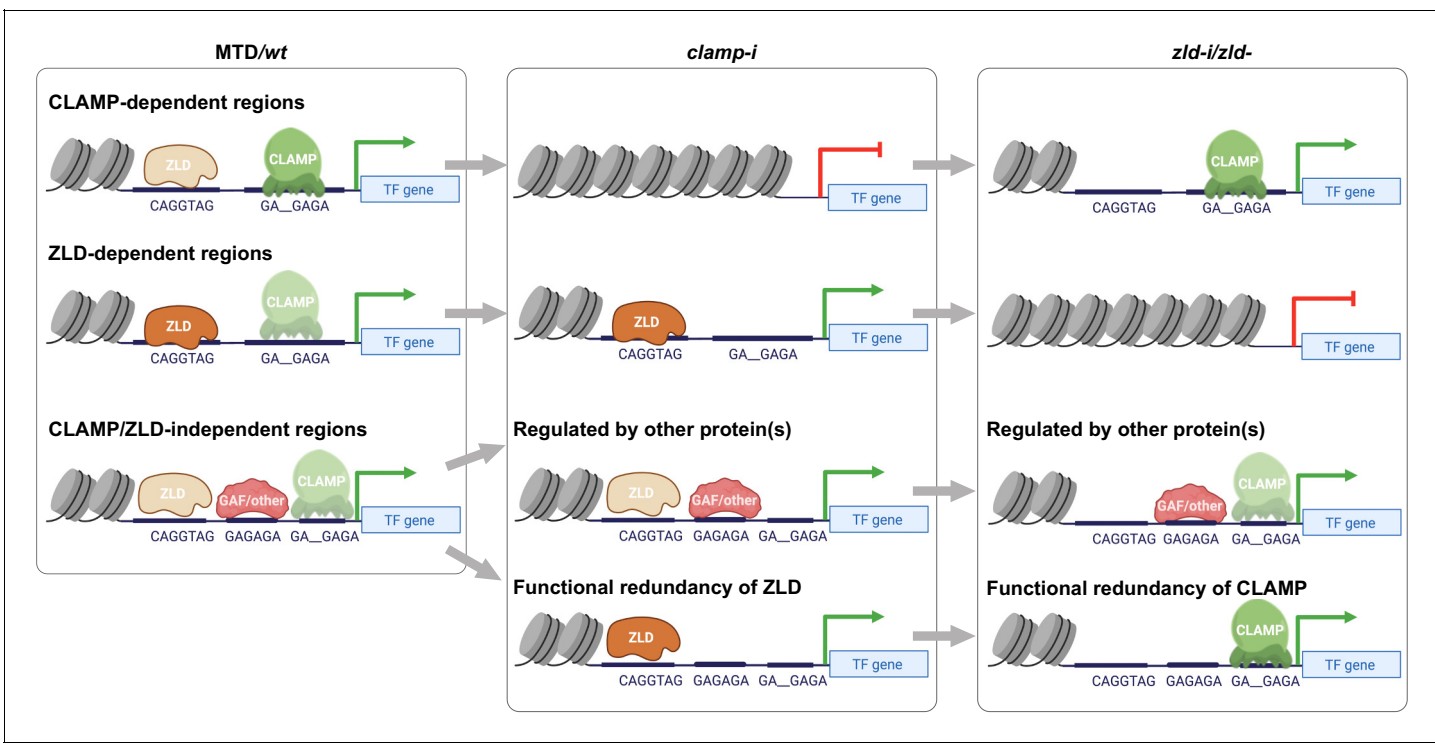

**Figure 6.** Model for how CLAMP and ZLD pioneer factor function together to define chromatin accessibility in early embryos. CLAMP and ZLD function together at promoters to regulate each other's occupancy and gene expression of genes encoding other key TFs. We defined CLAMP and ZLD co-bound peaks in early embryos, which revealed roles for CLAMP and ZLD in defining chromatin accessibility and activating zygotic transcription at a subset of the zygotic genome.CLAMP-dependent regions: CLAMP promotes ZLD enrichment at these sites where CLAMP binding increases chromatin accessibility and regulates target gene expression. These sites are closed and lack binding of ZLD when maternal *clamp* is depleted, and they remain open and transcription is activated when maternal *zld* is depleted. ZLD-dependent regions: ZLD modulates chromatin opening and transcription at these sites that are bound by CLAMP but do not depend on CLAMP for chromatin accessibility. These sites are closed and lack binding of CLAMP when maternal *zld* is depleted, and they remain open and active when maternal *clamp* is depleted. CLAMP/ZLD-independent regions: GAF or other TFs open chromatin at locations co-bound by CLAMP and ZLD where chromatin accessibility is not altered when each factor is depleted individually. CLAMP and ZLD could also function redundantly at some of these loci. These sites remain accessible and transcriptionally active upon either maternal *zld* or *clamp* depletion. CLAMP, chromatin-linked adaptor for male-specific lethal (MSL) proteins; TF, transcription factor.

chromatin accessibility at promoters throughout the genome, depletion of CLAMP alone mainly drives changes at promoters of specific genes that often encode TFs that are important for early development, consistent with phenotypic data. These findings indicate that CLAMP and ZLD regulate ZGA in different ways: ZLD mediates chromatin opening globally, while the CLAMP functions in a more targeted way at certain essential early TF genes. However, both proteins are critical to ZGA and loss of either is catastrophic in terms of overall embryonic development.

Moreover, ZLD binding and/or chromatin accessibility is not regulated by maternal depletion of CLAMP at all GA-rich sites in the genome. GAF is also enriched at these same ZLD-bound regions where ZLD is not required for chromatin accessibility (*Schulz et al., 2015*; *Gaskill et al., 2021*). Both CLAMP and GAF are deposited maternally (*Rieder et al., 2017*; *Hamm et al., 2017*) and bind to similar GA-rich motifs (*Kaye et al., 2018*). To test whether GAF compensates for the depletion of CLAMP or ZLD, we tried to perform GAF RNAi in the current study to prevent GAF from compensating for CLAMP depletion. However, we and other laboratories could not achieve depletion of GAF in early embryos by RNAi, likely due to autoregulation of its own promoter and its prion-like self-perpetuating function (*Tariq et al., 2013*).

We previously demonstrated that competition between CLAMP and GAF at GA-rich binding sites is essential for MSL complex recruitment in S2 cells (*Kaye et al., 2018*). Furthermore, CLAMP excludes GAF at the histone locus which co-regulates genes that encode the histone proteins (*Rieder et al., 2017*). However, we also observed synergistic binding between CLAMP and GAF at many additional binding sites (*Kaye et al., 2018*). The relationship between CLAMP and GAF in early embryos remains unclear. It is very possible that the competitive relationship has not been established in early embryos, since dosage compensation has not yet been initiated (*Prayitno et al., 2019*). Using GAF-dependent loci defined by *Gaskill et al., 2021*, we found that genomic loci where GAF functions largely overlap with regions where depletion of CLAMP or ZLD alone does not alter chromatin accessibility, indicating that GAF may function independently of CLAMP or ZLD or is functionally redundant. Future studies are required to distinguish between these models by examining how GAF and CLAMP affect each other's binding to co-bound loci and simultaneously eliminating both factors.

The GA-rich sequences targeted by CLAMP and GAF are distinct from each other in vivo and in vitro. GAF binding sites typically have 3.5 GA repeats; however, GAF is able to bind to as few as three bases (GAG) within the *hsp70* promoter and in vitro (*Wilkins and Lis, 1999*). In contrast, CLAMP binding sites contain an 8-bp core with a less well-conserved second GA dinucleotide within the core (GA__GAGA) (*Alekseyenko et al., 2008*). CLAMP binding sites also include a GAGAG pentamer at a lower frequency than GAF binding sites, and flanking bases surrounding the 8-bp core are critical for CLAMP binding (*Kaye et al., 2018*). Therefore, GAF and CLAMP may have overlapping and non-overlapping functions at different loci, tissues, or developmental stages. Moreover, another TF, Pipsqueak (Psq) also binds to sites containing the GAGAG motif, and has multiple functions during oogenesis and embryonic pattern formation and functions with Polycomb in three-dimensional genome organization (*Lehmann et al., 1998*; *Gutierrez-Perez et al., 2019*). In the future, an optogenetic inactivation approach could be used to remove CLAMP, GAF, and/or Psq simultaneously in a spatial and temporal manner (*McDaniel et al., 2019*).

## CLAMP and ZLD regulate each other's binding via their own motifs

ZLD is an essential TF that regulates activation of the first set of zygotic genes during the minor wave of ZGA and thousands of genes transcribed during the major wave of ZGA at nuclear cycle 14 (*Liang et al., 2008*; *Harrison et al., 2011*). ZLD also establishes and maintains chromatin accessibility of specific regions and facilitates TF binding and early gene expression (*Sun et al., 2015*; *Schulz et al., 2015*). CLAMP regulates histone gene expression (*Rieder et al., 2017*), X chromosome dosage compensation (*Soruco et al., 2013*), and establishes/maintains chromatin accessibility (*Urban et al., 2017b*). Nonetheless, it remained unclear whether and how CLAMP and ZLD functionally interact during ZGA. Here, we demonstrate that CLAMP and ZLD function together at a subset of promoters that often encode other transcriptional regulators.

ZLD regulates CLAMP occupancy earlier than CLAMP regulates ZLD occupancy. Genomic loci at which CLAMP is dependent on ZLD early (0–2 hr) in development often become independent from ZLD later (2–4 hr), with the caveat that ZLD depletion is not as effective later in development. Therefore, CLAMP may require the pioneering activity of ZLD to access specific loci before ZGA, but ZLD

may no longer be necessary once CLAMP binding is established. Also, our results suggest that CLAMP is a potent regulator of ZLD binding, especially in 2–4 hr embryos. ZLD can bind to many more promoter regions at 0–2 hr, while CLAMP mainly binds to introns early in development but occupies promoters later at 2–4 hr. Therefore, CLAMP may require ZLD to increase chromatin accessibility of these promoter regions (*Schulz et al., 2015*).

In addition to its role in embryonic development, CLAMP also plays an essential role in targeting the MSL male dosage compensation complex to the X chromosome (*Soruco et al., 2013*). *Drosophila* embryos initiate X chromosome counting in nuclear cycle 12 and start the sex determination cascade prior to the major wave of ZGA at nuclear cycle 14 (*Gergen, 1987*; *ten Bosch et al., 2006*). However, most dosage compensation is initiated much later in embryonic development (*Prayitno et al., 2019*). Our data support a model in which CLAMP functions early in the embryo prior to MSL complex assembly to open up specific chromatin regions for MSL complex recruitment (*Urban et al., 2017b*; *Rieder et al., 2019*). Moreover, ZLD likely functions primarily as an early pioneer factor, whereas CLAMP has pioneer functions in both early and late-ZGA embryos. Consistent with this hypothesis, CLAMP binding is enriched at both early and late zygotic genes. In contrast, ZLD binding binds more frequently to early genes, suggesting that there may be a sequential relationship between occupancy of these two TFs at some loci during early embryogenesis.

The different characteristics of dependent and independent CLAMP and ZLD binding sites also provide insight into how early TFs work together to regulate ZGA. At dependent sites, there are often relatively broad peaks of CLAMP and ZLD that are significantly enriched for clusters of motifs for the required protein. Our CLAMP gel shift assays and those previously reported (*Kaye et al., 2018*) also show multiple shifted bands consistent with possible multimerization. CLAMP contains two central disordered prion-like glutamine-rich regions (*Kaye et al., 2018*), a domain that is critical for transcriptional activation and multimerization in vivo in several TFs, including GAF (*Wilkins and Lis, 1999*). Moreover, glutamine-rich repeats alone can be sufficient to mediate stable protein multimerization in vitro (*Stott et al., 1995*). Therefore, it is reasonable to hypothesize that the CLAMP glutamine-rich domain also functions in CLAMP multimerization.

In contrast, ZLD fails to form dimers or multimers (*Hamm et al., 2015*; *Hamm et al., 2017*), indicating that ZLD most likely binds as a monomer. There is no evidence that CLAMP and ZLD have any direct protein-protein interaction at sites where they depend on each other to bind. For example, mass spectrometry results that identified dozens of CLAMP-associated proteins did not identify ZLD (*Urban et al., 2017b*). No data has validated any protein-protein interactions of ZLD with itself as a multimer or between ZLD and any other TFs (*Hamm et al., 2017*). In the future, simultaneous ablation of maternal CLAMP and ZLD will allow the analysis of potential functional redundancy at a subgroup of genomic loci. Our study suggests that regulating the chromatin landscape in early embryos to drive ZGA requires the function of multiple pioneer TFs.

## Materials and methods

### Recombinant protein expression and purification of CLAMP

MBP-tagged CLAMP DBD was expressed and purified as described previously (*Kaye et al., 2018*). MBP-tagged (pTHMT, *Peti and Page, 2007*) FL CLAMP protein was expressed in *Escherichia coli* BL21 Star (DE3) cells (Life Technologies). Bacterial cultures were grown to an optical density of 0.7–0.9 before induction with 1 mM isopropyl-β-D-1-thiogalactopyranoside (IPTG) for 4 hr at 37°C.

Cell pellets were harvested by centrifugation and stored at −80°C. Cell pellets were resuspended in 20 mM Tris, 1 M NaCl, 0.1 mM $ZnCl_2$, and 10 mM imidazole pH 8.0 with one EDTA-free protease inhibitor tablet (Roche) and lysed using an Emulsiflex C3 (Avestin). The lysate was cleared by centrifugation at 20,000 rpm for 50 min at 4°C, filtered using a 0.2 μm syringe filter, and loaded onto a HisTrap HP 5 ml column. The protein was eluted with a gradient from 10 to 300 mM imidazole in 20 mM Tris, 1.0 M NaCl pH 8.0, and 0.1 mM $ZnCl_2$. Fractions containing MBP-CLAMP FL were loaded onto a HiLoad 26/600 Superdex 200 pg column equilibrated in 20 mM Tris, 1.0 M NaCl, pH 8.0. Fractions containing FL CLAMP were identified by SDS-PAGE and concentrated using a centrifugation filter with a 10-kDa cutoff (Amicon, Millipore) and frozen as aliquots.

## In vitro assembly of nucleosomes

The 240 bp 5C2 DNA fragment used for nucleosome in vitro assembly was amplified from 276 bp 5C2 fragments (50 ng/µl, IDT gBlocks gene fragments) by PCR (see 276 bp 5C2 and primer sequences below) using OneTaq Hot Start 2× Master Mix (New England Biolabs). The DNA was purified using the PCR Clean-Up Kit (Qiagen) and concentrated to 1 µg/µl by SpeedVac Vacuum (Eppendorf). The nucleosomes were assembled using the EpiMark Nucleosome Assembly Kit (New England Biolabs) following the kit's protocol.

5C2 (276 bp), **bold** sequences are CLAMP-binding motifs, underlined sequences are primer binding sequences:

TCGACGACTAGTTTAAAGTTATTGTAGTTCTTAGAGCAGAATGTATTTTAAATATCAATG
TTTCGATGTAGAAATTGAATGGTTTAAATCACGTTCACACAACTTA**GAAAGAGATAG**C-
GATGGCGGTGT**GAAAGAGAGCGAGATAG**TTGGAAGCTTCATG**GAAATGAAAGAGAGG
TAG**TTTTTGGAAATGAAAGTTGTACTAGAAATAAGTATTTTATGTATATAGAATATCGAAG
TACAGAAATTCGAAGCGATCTCAACTTGAATATTATATCG

Primers for 5C2 region (product is 240 bp):

Forward: TTGTAGTTCTTAGAGCAGAATGT
Reverse: GTTGAGATCGCTTCGAATTT

## Electrophoretic mobility shift assays

DNA or nucleosome probes at 35 nM (700 fmol/reaction) were incubated with MBP-tagged CLAMP DBD protein or MBP-tagged FL CLAMP protein in a binding buffer. The binding reaction buffer conditions are similar to conditions previously used to test ZLD nucleosome binding (*McDaniel et al., 2019*) in 20 µl total volume: 7.5 µl BSA/HEGK buffer (12.5 mM HEPES, pH 7.0, 0.5 mM EDTA, 0.5 mM EGTA, 5% glycerol, 50 mM KCl, 0.05 mg/ml BSA, 0.2 mM PMSF, 1 mM DTT, 0.25 mM $ZnCl_2$, and 0.006% NP-40) 10 µl probe mix (5 ng poly[d-(IC)], 5 mM $MgCl_2$, 700 fmol probe), and 2.5 µl protein dilution (0.5µM, 1 µM, and 2.5 µM) at room temperature for 60 min. Reactions were loaded onto 6% DNA retardation gels (Thermo Fisher Scientific) and run in 0.5× Tris–borate–EDTA buffer for 2 hr. Gels were post stained with GelRed Nucleic Acid Stain (Thermo Fisher Scientific) for 30 min and visualized using the ChemiDoc MP imaging system (Bio-Rad).

## Fly stocks and crosses

To deplete maternally deposited *clamp* or *zld* mRNA throughout oogenesis, we crossed a maternal triple driver (MTD-GAL4, Bloomington, #31777) line (*Ni et al., 2011*) with a Transgenic RNAi Project (TRiP) *clamp* RNAi line (Bloomington, #57008), a TRiP *zld* RNAi line (from C. Rushlow lab) or *egfp* RNAi line (Bloomington, #41552). The *egfp* RNAi line was used as control in smFISH immunostaining and imaging experiments. The MTD-GAL4 line alone was used as the control line in ATAC-seq and ChIP-seq experiments.

Briefly, the MTD-GAL4 virgin females (5–7 days old) were mated with TRiP UAS-RNAi males to obtain MTD-Gal4/UAS-RNAi line daughters. The MTD drives RNAi during oogenesis in these daughters. Therefore, the targeted mRNA is depleted in their eggs. Then MTD-Gal4/UAS-RNAi daughters were mated with males to produce embryos with depleted maternal *clamp* or *zld* mRNA and used for ATAC-seq and ChIP-seq experiments. The embryonic phenotypes of the maternal *zld*⁻ TRiP RNAi line were confirmed previously (*Sun et al., 2015*). Maternal *clamp*⁻ embryonic phenotypes of the TRiP *clamp* RNAi line were confirmed by immunofluorescent staining in our study. Moreover, we validated CLAMP or ZLD protein knockdown in early embryos by Western blotting using the Western Breeze Kit (Invitrogen) and measured *clamp* and *zld* mRNA levels by qRT-PCR (*Figure 1—figure supplement 1B,C* and *Figure 1—source data 1*).

## Embryo collections

To optimize egg collections, young (5–7 days old) females and males were mated. To ensure mothers do not lay older embryos during collections, we first starved flies for 2 hr in the empty cages and discarded the first 2 hr grape agar plates with yeast paste (Plate set #0). When we collected eggs for the experiments, we put flies in the cages with grape agar plates (Plate set #1) with yeast paste for egg laying for 2 hr. Then, we replaced Plate set #1 with a new set of plates (Plate set #2) at the 2 hr

time point. We kept Plate set #1 embryos (without any adult flies) to further develop for another 2 hr to obtain 2–4 hr embryos. At the same time, we obtained newly laid 0–2 hr embryos from Plate set #2. Therefore, this strategy successfully prevented cross-contamination between 0–2 hr (Plate set #2) and 2–4 hr embryos (Plate set #1).

## smFISH, Immunostaining and Imaging

For whole embryo single-molecule fluorescence in situ hybridization (smFISH) and immunostaining and subsequent imaging, standard protocols were used (*Little and Gregor, 2018*). smFISH probes complementary to *run* were a gift from Thomas Gregor, and those complementary to *eve* were a gift from Shawn Little. The concentrations of the different dyes and antibodies were as follows: Hoechst (Invitrogen, 3 µg/ml), anti-NRT (Developmental Studies Hybridoma Bank BP106, 1:10), AlexaFluor secondary antibodies (Invitrogen Molecular Probes, 1:1000). Imaging was done using a Nikon A1 point-scanning confocal microscope with a 40× oil objective. Image processing and intensity measurements were done using ImageJ software (NIH). Figures were assembled using Adobe Photoshop CS4.

## ATAC-seq in embryos

We conducted ATAC-seq following the protocol from *Blythe and Wieschaus, 2016*. 0–2 hr or 2–4 hr embryos were laid on grape agar plates, dechorionated by 1 min exposure to 6% bleach (Clorox) and then washed three times in deionized water. We homogenized 10 embryos and lysed them in 50 µl lysis buffer (10 mM Tris 7.5, 10 mM NaCl, 3 mM $MgCl_2$, and 0.1% NP-40). We collected nuclei by centrifuging at 500$g$ at 4°C and resuspended nuclei in 5 µl TD buffer with 2.5 µl Tn5 enzyme (Illumina Tagment DNA TDE1 Enzyme and Buffer Kits). We incubated samples at 37°C for 30 min at 800 rpm (Eppendorf Thermomixer) for fragmentation, and then purified samples with Qiagen MinElute columns before PCR amplification. We amplified libraries by adding 10 µl DNA to 25 µl NEBNext HiFi 2× PCR mix (New England Biolabs) and 2.5 µl of a 25 µM solution of each of the Ad1 and Ad2 primers. We used 13 PCR cycles to amplify samples from 0 to 2 hr embryos and 12 PCR cycles to amplify samples from 2 to 4 hr embryos. Next, we purified libraries with 1.2× Ampure SPRI beads. We performed three biological replicates for each genotype (n=2) and time point (n=2). We measured the concentrations of 12 ATAC-seq libraries by Qubit and determined library quality by Bioanalyzer. We sequenced libraries on an Illumina Hi-seq 4000 sequencer at GeneWiz (South Plainfield, NJ) using the 2 × 150 bp mode. ATAC-seq data is deposited at NCBI GEO and the accession number is GSE152596.

## ChIP-seq in embryos

We performed ChIP-seq as previously described (*Blythe and Wieschaus, 2015*). We collected and fixed ~100 embryos from each MTD-GAL4 and RNAi cross 0–2 hr or 2–4 hr after egg lay. We used 3 µl of rabbit anti-CLAMP (*Soruco et al., 2013*) and 2 µl rat anti-ZLD (from C. Rushlow lab) per sample. We performed three biological ChIP replicates for each protein (n=2), genotype (n=3), and time point (n=2). In total, we prepared 36 libraries using the NEBNext Ultra ChIP-seq Kit (New England Biolabs) and sequenced libraries on the Illumina HiSeq 2500 sequencer using the 2 × 150 bp mode. ChIP-seq data is deposited at NCBI GEO and the accession number is GSE152598.

## Computational analyses

### ATAC-seq analysis

Prior to sequencing, the Fragment Analyzer showed the library top peaks were in the 180–190 bp range, which is comparable to the previously established embryo ATAC-seq protocol (*Haines, 2017*). Demultiplexed reads were trimmed of adapters using TrimGalore (*Krueger, 2017*) and mapped to the *Drosophila* genome dm6 version using Bowtie2 (v. 2.3.0) with option `-very-sensitive, -no-mixed, -no-discordant, -dovetail -X 2000 k 2`. We used Picard tools (v. 2.9.2) and SAMtools (v.1.9, *Li et al., 2009*) to remove the reads that were unmapped, failed primary alignment, or duplicated (-F 1804), and retain properly paired reads (-f 2) with MAPQ >30. After quality trimming and mapping, the Picard tool reported the mean fragment sizes for all ATAC-seq mapped reads are between 125 and 161 bp. As expected, we observed three classes of peaks: (1) a sharp peak at

<100 bp (open chromatin); (2) a peak at ~200 bp (mono-nucleosome); and (3) other larger peaks (multi-nucleosomes).

After mapping, we used Samtools to select a fragment size ≤100 bp within the bam files to focus on open chromatin. Peak regions for open chromatin regions were called using MACS2 (v. 2.1.1, *Zhang et al., 2008*) with parameters -f BAMPE -g dm `--call-summits`. ENCODE blacklist was used to filter out problematic regions in dm6 (*Amemiya et al., 2019*). Bam files and peak bed files were used in DiffBind v.3.12 (*Stark and Brown, 2019*) for count reads (dba.count), library size normalization (dba.normalize), and calling (dba.contrast) DA region with the DESeq2 method. Peak regions (201 bp) were centered by peak summits and extended 100 bp on each side. Sites were defined as DA with statistically significant differences between conditions using absolute cutoffs of FC>0.5 and FDR<0.1. We report all accessible peaks from DiffBind in *Figure 2—source data 1*.

We used DeepTools (v. 3.1.0, *Ramírez et al., 2014*) to generate enrichment heatmaps (CPM normalization), and average profiles were generated in DeepStats (*Gautier, 2020*). We used 1× depth (reads per genome coverage, RPGC) normalization in Deeptools bamCoverage for making the coverage Bigwig files and uploaded to IGV (*Robinson et al., 2011*) for genomic track visualizations. Homer (v. 4.11, *Givler and Lilienthal, 2005*) was used for de novo motif searches. Visualizations and statistical tests were conducted in *R Development Core Team, 2014*. Specifically, we annotated peaks to their genomic regions using R packages Chipseeker (*Yu et al., 2015*) and we performed gene ontology enrichment analysis using clusterProfiler (*Yu et al., 2012*). Boxplot and violin plots were generated using ggplot2 (*Wickham, 2009*) package.

## ChIP-seq analysis

Briefly, we trimmed ChIP-seq raw reads with TrimGalore (*Krueger, 2017*) with a minimal phred score of 20, 36 bp minimal read length, and Illumina adaptor removal. We then mapped cleaned reads to the *D. melanogaster* genome (UCSC dm6) with Bowtie2 (v. 2.3.0) with the –very-sensitive-local flag feature. We used Picard tools (v. 2.9.2) and SAMtools (v. 1.9, *Li et al., 2009*) to remove the PCR duplicates. We used MACS2 (v. 2.1.1, *Zhang et al., 2008*) to identify peaks with default parameters and MSPC (v. 4.0.0, *Jalili et al., 2015*) to obtain consensus peaks from three replicates. The peak number for each sample was summarized in *Table 1*. ENCODE blacklist was used to filter out problematic regions in dm6 (*Amemiya et al., 2019*). We identified DB and non-DB between MTD and RNAi samples using DiffBind (v. 3.10, *Stark and Brown, 2019*) with the DESeq2 method. Peak regions (501 bp) were centered by peak summits and extended 250 bp on each side. The DB and non-DB peak numbers are summarized in *Table 1*. DB was defined with absolute FC>0.5 and FDR<0.05 (*Table 1—source data 1*).

We used DeepTools (v. 3.1.0, *Ramírez et al., 2014*) to generate enrichment heatmaps and average profiles. Bigwig files were generated with DeepTools bamCompare (scale factor method: SES; Normalization: log$_2$) and uploaded to IGV (*Robinson et al., 2011*) for genomic track visualization. We used Homer (v. 4.11, *Givler and Lilienthal, 2005*) for de novo motif searches and genomic annotation. Intervene (*Khan and Mathelier, 2017*) was used for intersection and visualization of multiple peak region sets. Visualizations and statistical tests were conducted in *R Development Core Team, 2014*. Specifically, we annotated peaks to their genomic regions using the R package Chipseeker (*Yu et al., 2015*) and we did gene ontology enrichment analysis using clusterProfiler (*Yu et al., 2012*). Boxplots and violin plots were generated using the ggplot2 (*Wickham, 2009*) package.

## ATAC-seq and ChIP-seq data integration

We used Bedtools (*Quinlan and Hall, 2010*) intersection tool to intersect peaks in CLAMP ChIP-seq binding regions with CLAMP DA or non-DA peaks. Based on the intersection of the peaks, we defined four types of CLAMP related peaks: (1) DA with CLAMP, (2) DA without CLAMP, (3) non-DA with CLAMP, and (4) non-DA without CLAMP. Similarly, we defined ZLD related peaks by intersecting ZLD DA or non-DA peaks and ATAC-seq data sets (*Hannon et al., 2017*; *Soluri et al., 2020*) from wt and *zld* germline clone (*zld-*) embryos at the NC14 +12 min stage. Specifically, we defined four classes of genomic loci for ZLD-related classes: (1) DA with ZLD, (2) DA without ZLD, (3) non-DA with ZLD, and (4) non-DA, without ZLD. We used DeepTools (v. 3.1.0, *Ramírez et al., 2014*) to generate enrichment heatmaps for each subclass of peaks. Peaks locations in each CLAMP or ZLD-related category were summarized in *Table 2—source data 1*.

## ATAC-seq and RNA-seq data integration

We annotated genes near differential (down-DA) ATAC-seq peaks in R using detailRanges function from the csaw package (*Lun and Smyth, 2016*). Then we plotted the expression of genes using previously published RNA-seq data (*Rieder et al., 2017*).

## ChIP-seq and RNA-seq data integration

To define strong, weak, and unbound genes close to peaks in CLAMP or ZLD ChIP-seq data, we used the peak binding score reported in MACS2 -log10(p-value) of 100 as a cutoff value. We defined the following categories: (1) strong binding peaks: score greater than 100; (2) weak binding peak: score lesser than 100; (3) unbound peaks: the rest of the peaks that are neither strong or weak. Then, we annotated all peaks using Homer annotatePeaks (v. 4.11, *Givler and Lilienthal, 2005*). We then obtained the $\log_2$ fold change (*clamp-i*/MTD or *zld-i/yw*) of gene expression in the RNA-seq data set for each protein binding group: CLAMP (*Rieder et al., 2017*) or ZLD (*Schulz et al., 2015*). Boxplots and violin plots were generated using the ggplot2 (*Wickham, 2009*) package.

## Data sets

RNA-seq data sets from wt and maternal *clamp* depletion by RNAi were from GSE102922 (*Rieder et al., 2017*). RNA-seq data sets from *yw* wt and *zld* maternal RNAi were from GSE65837 (*Schulz et al., 2015*). ATAC-seq data from wt and *zld* germline clones were from GSE86966 (*Hannon et al., 2017*). Processed ATAC-seq data identifying differential peaks between wt and *zld* germline mutations were from *Soluri et al., 2020*.

# Acknowledgements

The authors thank Dr. Melissa Harrison, Tyler Gibson, and Marissa Gaskill for sending the GAF-dependent region bed file and helpful discussions. The authors thank members in the Larschan lab for feedback and discussions. This work was supported by NIH Grant F32GM109663, K99HD092625, and R00HD092625 to Dr. Leila Rieder and R35GM126994 to Dr. Erica Larschan, and in part by NSF Grant 1845734 and NIH Grant R01GM118530 to Dr. Nicolas L Fawzi.

# Additional information

### Funding

| Funder | Grant reference number | Author |
| --- | --- | --- |
| National Institute of General Medical Sciences | F32GM109663 | Leila Rieder |
| National Institute of General Medical Sciences | K99HD092625 | Leila Rieder |
| National Institute of General Medical Sciences | R00HD092625 | Leila Rieder |
| National Institute of General Medical Sciences | R35GM126994 | Erica Larschan |
| National Science Foundation | 1845734 | Nicolas Fawzi |
| National Institute of General Medical Sciences | R01GM118530 | Nicolas Fawzi |

The funders had no role in study design, data collection and interpretation, or the decision to submit the work for publication.

### Author contributions

Jingyue Duan, Conceptualization, Data curation, Formal analysis, Supervision, Validation, Investigation, Visualization, Methodology, Writing - original draft, Writing - review and editing; Leila Rieder, Conceptualization, Data curation, Software, Formal analysis, Supervision, Validation, Investigation, Visualization, Methodology, Writing - review and editing; Megan M Colonnetta, Visualization,

Methodology, Writing - review and editing; Annie Huang, Mary Mckenney, Validation; Scott Watters, Supervision, Validation, Visualization; Girish Deshpande, Supervision, Validation, Visualization, Writing - review and editing; William Jordan, Resources, Methodology; Nicolas Fawzi, Supervision, Funding acquisition, Methodology; Erica Larschan, Conceptualization, Resources, Data curation, Software, Formal analysis, Supervision, Funding acquisition, Validation, Investigation, Visualization, Methodology, Writing - original draft, Project administration, Writing - review and editing

### Author ORCIDs
Jingyue Duan (iD) https://orcid.org/0000-0001-6416-2250
Megan M Colonnetta (iD) http://orcid.org/0000-0001-5685-1670
Erica Larschan (iD) https://orcid.org/0000-0003-2484-4921

### Decision letter and Author response
Decision letter https://doi.org/10.7554/eLife.69937.sa1
Author response https://doi.org/10.7554/eLife.69937.sa2

## Additional files
### Supplementary files
• Transparent reporting form

### Data availability
Sequencing data have been deposited in GEO under accession code GSE152613.

The following dataset was generated:

| Author(s) | Year | Dataset title | Dataset URL | Database and Identifier |
|---|---|---|---|---|
| Rieder L, Colonnetta MM, Huang A, Mckenney M, Watters S, Deshpande G, Jordan W, Fawzi N, Larschan E | 2020 | CLAMP and Zelda function together as pioneer transcription factors to promote Drosophila zygotic genome activation | https://www.ncbi.nlm.nih.gov/geo/query/acc.cgi?&acc=GSE152613 | NCBI Gene Expression Omnibus, GSE152613 |

The following previously published datasets were used:

| Author(s) | Year | Dataset title | Dataset URL | Database and Identifier |
|---|---|---|---|---|
| Rieder LE, Koreski KP, Boltz KA, Kuzu G, Urban JA, Bowman S, Zeidman A, Jordan WT, Tolstorukov MY, Marzluff WF, Duronio RJ, Larschan EN | 2019 | Histone locus regulation by the Drosophila dosage compensation adaptor protein CLAMP | https://www.ncbi.nlm.nih.gov/geo/query/acc.cgi?acc=GSE102922 | NCBI Gene Expression Omnibus, GSE102922 |
| Rieder L | 2015 | Zelda determines chromatin accessibility during the Drosophila maternal-to-zygotic transition | https://www.ncbi.nlm.nih.gov/geo/query/acc.cgi?acc=GSE65837 | NCBI Gene Expression Omnibus, GSE65837 |
| Rieder L | 2017 | Concentration dependent binding states of the Bicoid Homeodomain Protein | https://www.ncbi.nlm.nih.gov/geo/query/acc.cgi?acc=GSE86966 | NCBI Gene Expression Omnibus, GSE86966 |

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
