## [Decision Letter]

**Acceptance summary:**

The authors showed that CLAMP functions as a pioneer factor during zygotic genome activation in *Drosophila* on specific subset of target genes, particularly at target genes with GA-rich motifs that are not regulated by a well-established pioneer factor Zelda.

**Decision letter after peer review:**

[Editors’ note: the authors submitted for reconsideration following the decision after peer review. What follows is the decision letter after the first round of review.]

Thank you for submitting your work entitled "CLAMP and Zelda function together as pioneer transcription factors to promote *Drosophila* zygotic genome activation" for consideration by *eLife*. Your article has been reviewed by 3 peer reviewers, and the evaluation has been overseen by a Reviewing Editor and a Senior Editor. The reviewers have opted to remain anonymous.

Our decision has been reached after consultation between the reviewers. The manuscript by Duan and Rieder et al. describes, for the first time, how the CLAMP transcription factor acts as a pioneer TF in the fly embryo. They demonstrate that CLAMP can bind to nucleosome-bound DNA and that it binds to and generates accessible chromatin at a set of gene promoters in the early embryo, and that without this activity these genes fail to be transcribed during ZGA. They further describe fascinating cooperativity between CLAMP and ZLD, a previously identified pioneer TF in the fly embryo. The work will be of broad interest to both the developmental biology and transcription biology fields.

All the reviewers appreciated the importance and the potential impact of the work, but all raised issues with experimental approaches particularly regarding the following issues.

1. Bioinformatics analysis

2. Writing – all noted that the manuscript was not written well to allow readers to follow the logic easily. Sometimes, the authors appeared to make contradictory statements

Details can be found in individual reviews.

In the light of *eLife*'s policy to invite revisions only when revision experiments are unlikely to change major conclusion of the paper, we decided that this manuscript must be rejected at this point. However, we would like to note that all the reviewers are quite enthusiastic about this manuscript, and if you can address concerns, we would be ready to review the revised manuscript as a new submission, which will be handled by the same set of editors/reviewers.

*Reviewer #1:*

In this manuscript, Duan, Rieder and collaborators examine the potential role of the transcription factor (TF) CLAMP as a pioneer transcription factor and potential co-regulator of zygotic genome activation (ZGA) in *Drosophila*. First the authors perform an in vitro characterisation of the biochemical properties of CLAMP with respect to being able to bind nucleosomal DNA and the effect of the depletion of the protein in regulating transcription. The authors then perform ATAC-seq, and ChIP-seq in embryos depleted of CLAMP or Zelda (a known pioneer transcription factor in *Drosophila*, which is a master regulator of ZGA) to examine their binding patterns in relation to each other and how these affect chromatin accessibility during early embryonic development before (0-2 hours post fertilisation (hpf)) and after (2-4 hpf) ZGA.

The question is of interest since previous reports have described an enrichment of a GA-rich motif similar to that can be recognised by CLAMP in open chromatin at developmental time points around ZGA, posing the question as to whether CLAMP might be involved in this developmental transition.

Unfortunately, I find the manuscript very densely written and very difficult to follow. It has taken me much longer than anticipated to review the work since the concepts are not clearly introduced, and the description and interpretation of the data are confusing. Despite the dense writing, the manuscript lacks critical information regarding how analyses were done that preclude from a proper evaluation of the conclusions. For the parts of the manuscript where this is possible, it is unclear whether some of the analytical strategies are the most adequate to address the question, and whether the data supports the authors' conclusions.

1. The manuscript contains no information regarding the phenotypic effect of CLAMP depletion (or of Zelda depletion, which would serve as control). Are these embryos healthy? Do they stop developing? Are they viable at least until ZGA? How pure are the embryo collections? What is the proportion of embryos further than nuclear cycle 13,14 in the 0-2 hpf collections? In addition, the validation of the knockdowns is buried in Figure 3 —figure supplement 1C,D, and is not convincing. This is a critical piece of information which is necessary to interpret the results for all remaining figures in the manuscript.

2. The authors claim that CLAMP is a pioneer transcription factor that directly activates zygotically-transcribed genes (line 140), but in my opinion, this is not demonstrated in a convincing manner. First, the western blot presented in Figure 3 —figure supplement 1D doesn't allow for an evaluation of the presence of Zld at the protein level in CLAMP depleted embryos (see point 1 above). Furthermore, the analysis presented in Figure 1 —figure supplement 1A, shows that there is a widespread change in expression for zygotically expressed genes upon CLAMP depletion irrespective of the level of CLAMP binding, suggesting that overall the observed changes correspond to indirect effects.

3. The manuscript doesn't contain critical information regarding how the ATAC-seq data were analysed. Did the authors use only short/long/all fragments for their analyses? This is important since one can then evaluate whether the authors are measuring open chromatin (short fragments) or nucleosome positioning (long fragments). In addition, how are differentially accessible regions calculated? Is the genome binned? If so, how big are these bins? I am not sure that I fully understand how this analysis is done, and I don't know what "log concentration" refers to in Figure 2A, but I find difficult to reconcile the close to zero correlation values reported for the ATAC-seq datasets in the MTD and CLAMP depleted samples when the majority of dots in panel 2A are not different between the two conditions.

4. The genomic track plot presented in Figure 2B doesn't help in the interpretation of the data. First, the panel is missing the genomic coordinates, so one cannot determine what are the peaks that are presented in the figure. Second the panel shows multiple CLAMP binding sites that don't seem to present an open chromatin signature, suggesting that only a fraction of the binding is related to open chromatin in the control samples. It would be interesting, and, in my opinion, much more straightforward to test, how much of the binding that leads to open chromatin (irrespective of whether this would be scored as differential accessible regions, which might be confounded for the reasons outlined in points 1 and 2 above) occurs together, for example, with Zld binding.

5. I find the scatterplots and the regression analysis presented in Figure 2E very unconvincing. Although statistically significant, the correlations are extremely weak, and if the authors would compute R^2 this would be extremely low, highlighting that the majority of the variance observed in these measurements remains unaccounted for. So overall, in my opinion, most of the authors' conclusions in Figure 2 are not supported without further analysis.

6. I find the analytical strategy in Figure 3A-C inappropriate and difficult to understand. First, the authors use average plots that contain order of magnitude different number of regions across the comparisons. The manuscript does not contain specific details regarding how these plots are produced, but these can be severely affected by outliers, especially in the cases when the number of regions is low. Second, the author define four classes of regions, depending whether they are bound or not bound by the factor and differentially accessible, and a control region (not bound and not differentially accessible). The split in these classes is very confusing and difficult to follow throughout the rest of the manuscript. In my opinion, this might be easier to follow if the data were presented as heatmaps including the common set of regions then split into different classes. Of note, the authors use comparisons across the average plots to "validate" (line 231) or "confirm" (line 243) their analysis. I disagree that this is validation because the groups have been chosen presumably based on thresholding the same data, and a validation could only come from orthogonal data, which is not used here. If the authors would want to use average plots, these should at least include a sharing area representing a confidence interval. Only one of the lines in Figure 6D shows such a shading, although that this represents is not explained. This significantly affects the majority of analyses and interpretations in Figures, 3,4,5,6.

7. There is a significant difference in the shape of the average profiles for the ATAC-seq data for the fourth group in both comparisons in Figure 3A. This is meant to be the control group in both cases, of non-DA, non-bound peaks. Since the controls are qualitative different between the CLAMP and Zelda depleted experiments, I wonder whether strong conclusions can be obtained from the comparisons of the other average profiles. In this respect, the p-values in Table 1 need to be corrected for multiple testing. Under Bonferroni correction, the p-value of the overlap between DA Zld-bound and DA CLAMP-bound is not significant, which is in disagreement with the authors' conclusions (line 247-248). This also affects the motivation for the analysis in Figure 4 (lines 312-313).

8. Figure 3D is misleading since the scales in both plots are different and do not let the reader appreciate the extent of the changes for the different classes. It might help with the visualisation if the authors would use a violin plot, or if they would plot the data without the outliers and including notches for the boxes. In my opinion, the results in Figure 3D indicate that there is a downregulation of gene expression at CLAMP-bound genes irrespective of changes in chromatin accessibility. This would challenge the authors' own conclusions of a pioneer role for CLAMP at those sites that don't change accessibility. It is therefore unclear how the authors arrive at the conclusions in lines 294-296.

9. Does the list of motifs included in Figure 4A contain all the set of significant motifs found in these regions? Without this information, it is not possible to evaluate the statement in lines 319-320. Furthermore, this can be due to the fact that there is a significantly lower number of DA CLAMP-bound in 2-4 hours compared with the 0-2 hours data, which might affect the significance of these enrichments. In my opinion, a better alternative would be to show the number of Zelda events that occur in these regions. Without this information, the conclusion stated in this paragraph (lines 320-322) is not supported.

10. The authors state that the majority of Zelda binding sites are not affected upon CLAMP depletion (line 345). However, I find this statement puzzling, since Figure 4 —figure supplement 1A shows a significant effect in the level of Zelda binding upon CLAMP KD throughout most of the regions shown in the heatmap.

11. Related to the point above, it would be useful if the authors could clarify how the data presented in Figure 4 —figure supplement 1A and Figure 4 —figure supplement 1D are different. Also, please note that this figure is missing the labelling of the x-axis. This is also the case for Figure 4 —figure supplement 1E.

12. The conclusions reached by the authors in lines 354-356 regarding the effect of Zld in CLAMP binding are not supported since, as the authors acknowledge, the experimental design is confounded by the up-regulation of Zld in the 2-4 hours time point.

13. I am unable to understand the interpretation of the results presented in Figure 4F and 4G. In any case, the results in Figure 4G might be confounded by the increased in Zld expression at 2-4 hours, as mentioned by the authors before.

14. Figure 5A lacks genomic coordinates, which makes it impossible to interpret the plot. In addition, the scales of the signal are also not readable, which makes it impossible to evaluate the robustness of the binding represented and the comparison. Related to this figure, does the difference in peak size depend on the number of individual binding events? I am unable to follow the results presented in Figure 5 —figure supplement 1E,F, and the interpretation of the results of Figure 5E.

15. I find it difficult to understand the statement in lines 458-459 because I do not understand what is the nature of the interdependent relationship between Zld and CLAMP binding to chromatin.

16. It is unclear whether the data in Figure 6C refers to the dependent or independent sites, since both seem to gain accessibility upon Zld depletion. I find this observation difficult to reconcile with the results presented in Figure 4 —figure supplement 1A,E that suggest that Zld depletion leads to an overall reduction of CLAMP binding. I would have expected then to observe a loss of accessibility, but not a gain. How do the authors explain this puzzling observation? In any case, since the gain in accessibility seems to be independent of CLAMP binding, since it occurs in both groups, can the authors be confident that this is biological and not due to technical differences between the libraries? What is the overlap between the sites that are reported here and those reported gaining accessibility in Schulz et al.?

17. The results in Figure 6D are also very difficult to interpret, especially given the limited effect of Zld depletion on CLAMP binding at 2-4 hours. How do the authors explain these results?

18. I don't understand the authors reasoning for the statement in lines 490-492. The authors' own analysis shows that the overlap between the set of downregulated genes at 2-4 hours is not better than one could expect by random chance (Figure 6F). How do the authors then conclude that there is co-regulation for hundreds of genes after ZGA?

*Reviewer #2:*

In the current manuscript under review, Duan et al. address the question of the role of GA-repeat binding factor CLAMP on the process of ZGA. The question of ZGA, particularly that of which pioneer factors establish patterns of chromatin accessibility and promote the expression of the first zygotic transcripts has received heavy attention in recent years. Notably, although another pioneer, Zelda, has a critical role for driving ZGA for a subset of zygotic genes by several measures, the vast majority of genomic locations either require a combination of Zelda and another factor, or another factor entirely. Several prior studies have pointed to enrichment for a GA-repeat motif within this class of sites. Identifying and characterizing the role of such a second maternal pioneer would represent a significant advance for the field as well as more broadly across biological fields as the question of pioneering touches on several key aspects of transcriptional regulation and epigenetics.

While Duan et al. present data that (1) CLAMP binds its motif even in the nucleosome-associated state; (2) CLAMP loss of function leads to some amount of reduced chromatin accessibility; (3) Some CLAMP and Zld DNA binding is interdependent; (4) Loss of CLAMP function affects gene transcription-- the manuscript in its current state is far from suitable for publication. My primary concern is that the data presentation of the genomics studies is extremely difficult to follow, that supporting data tables are either incompletely annotated or missing, in many cases it is nearly impossible to read the plot labels in the figures, and that the biological significance of the observations is not fully substantiated. In addition, certain controls have not been provided or even incorporated into experimental design. Also, there are issues with the presentation of the study, with factually incorrect statements and missing or unclear description of methods, and missing references (in some cases leading to factually incorrect statements).

This could be an important paper and it is therefore important that the presentation is watertight. I provide the comments below fully aware of current constraints on daily life, and in the spirit of wanting to minimize additional work for the authors. I think that overall the data already exists to improve the manuscript (or at least it should). But there is a fundamental question of whether the data are over-interpreted, and whether the effect of Clamp is as significant as the authors claim, at least within the framework of the process of ZGA.

1) The presentation of the genomics data analysis is very difficult to follow. I inspected the bigWig files for the ATAC data and had a hard time finding genomic regions where there is clear-cut evidence for CLAMP's role as a pioneer factor. Loading up the four ATAC conditions (two timepoints each control or clamp-i), as well as the Rieder CLAMP ChIP (NC14) and the 3h Harrison Zld ChIP-seq, I can find only a handful of regions where CLAMP has a clear all-or-nothing effect on chromatin accessibility, and these (few) sites are at regions where there is little Zld binding. These sites I did find by scrolling through nearly the entire genome are: 3' to CG11448, within iab-8, and possibly at the promoters of Vsx1 and 2. There are, however, numerous examples of regions where a 'differential enrichment' analysis could possibly yield a statistically significant difference between control and clamp-i, but there remains substantial accessible chromatin in the knockdown conditions. This latter phenomenon cannot be construed as evidence for pioneer activity, since it is expected that in the absence of the pioneer, the locus would be inaccessible. I am left with the question of whether the effect of Clamp on chromatin accessibility is oversold in this study.

The example regions plotted in the Figures also reveal potential issues in the analysis or interpretation of data: Figure 2B, CG11023: I had questions about what was going on in this plot which were cleared up by checking the bigWig files. For instance, I was curious why the light blue peak region indicator included regions with no ATAC signal in either control or clamp-i. Why also is there Clamp ChIP signal in this region? Upon inspection of the data, this plot shows base one of chr2L, and the blank region in the ATAC is presumably due (understandably) to mapping issues at the very telomeric end of the chromosome. Why is a peak called here? Why does the peak end within a peak of ATAC signal and not include this whole region? Significantly more concerning is that when I examine this region, my conclusion is that this whole region is likely very low signal that I would be reluctant to score both as "open" as well as "bound by Clamp". On the basis of this, I am reluctant to say that the bioinformatic analysis has been performed with sufficient rigor. Admittedly, this is based on one example image, but I would also point out that the authors have both only provided limited example regions, and have not provided a sufficiently documented 'peaks list' that includes regions that they feel are (1) bound by CLAMP, (2) bound by Zelda, (3) score as a member of the various groupings used to compare regions throughout the text (e.g. DA-Clamp bound, et cetera). The peaks list that the authors do provide is in a strange format and the column labels are not included in that file (nor can I find anywhere a description of that file, but I may have missed that in the submission materials). Nevertheless, it does not appear to indicate membership in any of the different classes from what I can tell.

It is similarly difficult to evaluate the conclusion that CLAMP has anything at all to do with ZGA (see below). Specifically, however, to the bioinformatics analysis: when RNAseq data is analyzed, is it limited to zygotic genes only (as defined either in DeRenzis 2007, or in the Li paper cited in the manuscript?), and is the magnitude of the effect large enough to warrant the conclusion that Clamp is required for ZGA? For comparison, loss of Zelda function results in near zero transcripts produced from a subset of zygotic genes (and corresponding elimination full stop of chromatin accessibility at those loci). I'm worried that the authors are placing too much weight on "significant" p-values without considering if the magnitude of the effect supports the stated conclusions. If the effect of clamp-i is minimal on transcription and chromatin accessibility, which it may be based on my limited examination of the raw data, I see no way to justify the conclusion that Clamp has any major role in ZGA.

I also have a difficult time finding any Zld-bound loci that convincingly show loss of accessibility in the clamp-i data.

*Reviewer #3:*

The manuscript submitted by Duan and Rieder et al. describes, for the first time, how the CLAMP transcription factor acts as a pioneer TF in the fly embryo. They demonstrate that CLAMP can bind to nucleosome-bound DNA and that it binds to and generates accessible chromatin at a set of gene promoters in the early embryo, and that without this activity these genes fail to be transcribed during ZGA. They further describe fascinating cooperativity between CLAMP and ZLD, a previously identified pioneer TF in the fly embryo. Their results are compelling and rigorous, and the work will be of broad interest to both the developmental biology and transcription biology fields.

One concern is in Figure 2E, This scatterplot and correlation is not particularly convincing. The fact that the positive correlation is very minor needs to be emphasized properly in the text. Moreover, in our opinion, there is a better way of doing this. ATAC-seq and RNA-seq are very different assays and as such it is to be expected that the fold change upon depletion of a factor should not be expected to be correlated in magnitude between assays, only the change in direction. The dynamic range of change you can expect in RNA-seq is much greater than that in ATAC-seq because mRNA is much more abundant than its cognate DNA for transcribed genes. We think the authors should simply display a Venn diagram of genes/promoters that move in the same direction, i.e. what fraction of the genes have the same directionality of change. We do not think that comparing the magnitude of these changes is particularly useful or informative in this case.

[Editors’ note: further revisions were suggested prior to acceptance, as described below.]

Thank you for submitting your work entitled "CLAMP and Zelda function together as pioneer transcription factors to promote *Drosophila* zygotic genome activation" for consideration by *eLife*. Your article has been reviewed by 3 peer reviewers, and the evaluation has been overseen by a Reviewing Editor and a Senior Editor. The reviewers have opted to remain anonymous.

We are sorry to say that, after consultation with the reviewers, we have decided that your work will not be considered further for publication by *eLife*. Although all the reviewers appreciated the improvement that the authors made since the last submission, reviewer #3 noticed major issues with the analysis of the sequencing data that affect the major conclusion. During consultation session among the reviewers and editors, others agreed that these are major issues, precluding the publication of your manuscript, at least in its current form. If these issues are addressed appropriately, we would be happy to re-consider the manuscript, but per *eLife*'s policy that revision only be invited when the major conclusion is unlikely to change as a result of revision, we are declining the current version of the work.

*Reviewer #1:*

The authors have satisfactorily addressed all our previous concerns, and the resubmitted manuscript is much better written and easier to understand than the previous submission. We feel the additional experiments the authors performed are sufficient to address concerns raised in the first round of review. This manuscript demonstrates clearly that CLAMP is acting as a pioneer factor in *Drosophila embryos*, and there is cooperativity between CLAMP and ZELDA at certain gene promoters.

*Reviewer #2:*

The authors have satisfactorily addressed all my questions. In my opinion the manuscript has vastly improved with the new data and analysis, and I do not have additional questions about the work.

*Reviewer #3:*

In the revised manuscript, Duan and colleagues have addressed some of the issues that were raised upon the original review. The authors have generally improved the presentation of their data and have rendered the results easier to interpret. However, despite these improvements, upon inspection of the differential enrichment analysis, the magnitude of effect of Clamp on differential chromatin accessibility is significantly overstated.

It is appreciated that the authors re-sequenced some of their lower depth samples for the resubmitted version. It is also appreciated that the authors have now provided the annotated tables from the differential enrichment analysis. In my original review, I mentioned that I manually searched nearly the entire genome while struggling to find more than a few examples of convincing loss of ATAC signal in the clamp-i data. I have now reviewed the differential enrichment analysis ("Table 4-source data 2", referred to as "Table 1 Source Data 2" in the text (line 681)) and note the following issue:

- The authors appear to have relied not on the FDR but rather on individual, independently calculated p-values for reckoning the number of differentially accessible peaks in the differential enrichment analysis. Table 1 reports 76 "Up", 1675 "Down", and 9465 "None" effects on accessibility in clamp-i embryos versus control. In the supplied source data, it is clear that the authors set a p-value cutoff of 0.05 for calculating these numbers. What isn't mentioned in the text is that this cutoff corresponds to a 32% FDR. Typically, a 5% FDR rate is chosen to minimize incorrect rejections of the null hypothesis that arise due to multiple testing. Using this standard, the total number of differential peaks in the 0-2 hour comparison is only 95, with 73 sites showing a reduction, and 22 showing an increase. Again, a manual inspection of a sampling of these regions shows marginal differences in magnitude between the few regions that do pass significance testing at 5% FDR.

Even fewer regions are differentially accessible in the 2-4 hour sample at a 5% FDR (total = 54, 33 "down", 21 "up").

On the basis of this observation, it would be hard to argue that CLAMP is playing a major role in regulating chromatin accessibility at ZGA. To me, this substantially casts doubt on the central premise of this manuscript and in fact suggests that CLAMP has only a minor effect on accessibility at this time.

[Editors' note: further revisions were suggested prior to acceptance, as described below.]

Thank you for resubmitting your work entitled "CLAMP and Zelda function together to promote *Drosophila* zygotic genome activation" for further consideration by eLife. Your revised article has been evaluated by Kevin Struhl (Senior Editor) and a Reviewing Editor.

The manuscript has been improved but there are some remaining issues that need to be addressed, as outlined below:

The authors have made a concerted effort to address the reviewer's concerns, and save for the remaining minor issues below, the manuscript is suitable for publication. While the reanalysis of the data has led to the conclusion that Clamp does not alter chromatin accessibility at as many sites in as non-redundant a way as Zelda, the work does document an interesting and critical interplay of pioneer transcription factors in early embryonic development, and it begins to understand the molecular underpinnings of that interplay. We think this work will be of broad interest and will help clarify how transcription factors act to establish chromatin accessibility and set-up the first steps in early embryonic transcription regulation.

1) Clamp as Pioneer: the authors have convincingly shown that Clamp binds to nucleosomal DNA using gel shift assays, and this result alone is probably sufficient to call it a pioneer factor in our view. However, the authors have also convincingly shown that the scope of Clamp pioneering accessibility of chromatin is very small compared to Zelda, but that like Zelda, loss of function is catastrophic in terms of overall development. Any use of "pioneer-like" can be replaced with "pioneer'. We also recommend that the authors carefully edit the Discussion to accurately describe the magnitude of Clamp's effect on accessibility, and to update the summation of results pending the outcome of points 2 and 3 below.

2) The reviewers agree that part of the new analysis presented in Figure 3 was not performed in an ideal manner to support the conclusions. The observation at line 245, for instance, is premature:

"Depletion of either maternal zld or clamp mRNA altered the genomic distribution of CLAMP and Zld: both factors shifted their occupancy from promoters to introns."

We request the authors either repeat this analysis more rigorously or eliminate the section entirely. The current analysis is performed by comparing independently called peak lists and placing emphasis on regions that are present or absent in each set. This approach is highly susceptible to thresholding artifacts associated with peak calling. All reviewers agree that a more rigorous approach would be either to perform this analysis on a single, union peak set followed by differential enrichment analysis, or coverage data between different treatments could be compared directly by generating XY-scatter plots of summed reads in each peak from a union peak list. If the conclusion of this section is correct, the genomic regions of interest should be significantly off the diagonal, and this can be statistically addressed.

3) The authors demonstrate that the knockdown efficiency of Zld RNAi is poor during the 2-4h timepoint (e.g. Figure 3, Fig. Supp. 1B). We caution the authors from drawing any strong conclusions about the effect of Zld on Clamp in the 2-4h time period. Please consider revising or eliminating the text beginning at line 262, where the weak effect of Zld on Clamp binding at 2-4 hours can possibly be attributed to incomplete knockdown.

4) For most of the heatmaps throughout the manuscript: the titles of the heatmaps incorrectly refer to "peaks", regardless of the data type presented in the heatmap. This can be confusing since the y-axis of the heatmap is some set of "peaks," and the data presented in the heatmap is ATAC-seq coverage or ChIP-seq coverage for a particular factor/genotype/timepoint. To improve readability, please revise heatmap plots to indicate the peak set on the y-axis, and relevant sample information in the header/title of each plot.

5) Paragraph beginning at line 341. Here, the authors are examining "gene expression changes caused by depleting maternal Zld at genes where CLAMP regulates Zld binding." The next sentence, however, talks about "genes where Zld regulates CLAMP binding." (Genes where "CLAMP regulates Zld binding" are never mentioned again.) This makes the logic of this paragraph difficult to interpret. Please revise.

6) In general, the reader has to work hard to clearly interpret the results section of this manuscript, particularly for Figures 4-5. Please consider editing the text related to Figures 4-5 for clarity.

---

## [Author Response]

[Editors’ note: the authors resubmitted a revised version of the paper for consideration. What follows is the authors’ response to the first round of review.]

Essential revisions:Reviewer #1:In this manuscript, Duan, Rieder and collaborators examine the potential role of the transcription factor (TF) CLAMP as a pioneer transcription factor and potential co-regulator of zygotic genome activation (ZGA) in *Drosophila*. First the authors perform an in vitro characterisation of the biochemical properties of CLAMP with respect to being able to bind nucleosomal DNA and the effect of the depletion of the protein in regulating transcription. The authors then perform ATAC-seq, and ChIP-seq in embryos depleted of CLAMP or Zelda (a known pioneer transcription factor in *Drosophila*, which is a master regulator of ZGA) to examine their binding patterns in relation to each other and how these affect chromatin accessibility during early embryonic development before (0-2 hours post fertilisation (hpf)) and after (2-4 hpf) ZGA.The question is of interest since previous reports have described an enrichment of a GA-rich motif similar to that can be recognised by CLAMP in open chromatin at developmental time points around ZGA, posing the question as to whether CLAMP might be involved in this developmental transition.Unfortunately, I find the manuscript very densely written and very difficult to follow. It has taken me much longer than anticipated to review the work since the concepts are not clearly introduced, and the description and interpretation of the data are confusing. Despite the dense writing, the manuscript lacks critical information regarding how analyses were done that preclude from a proper evaluation of the conclusions. For the parts of the manuscript where this is possible, it is unclear whether some of the analytical strategies are the most adequate to address the question, and whether the data supports the authors' conclusions.1. The manuscript contains no information regarding the phenotypic effect of CLAMP depletion (or of Zelda depletion, which would serve as control). Are these embryos healthy? Do they stop developing? Are they viable at least until ZGA?

We thank the reviewer for this important comment. We have performed new experiments to assess the phenotypic effect of *clamp* depletion. Overall, the phenotypes caused by the maternal depletion of *clamp* are very similar to those caused by the maternal depletion of *zld* including failed cellularization. Embryos stall their growth at the blastoderm stage when maternal *clamp* is depleted.

Furthermore, we performed smFISH or immunostaining to localize early patterning genes including *eve*, *run*, and Nrt before and after depletion of *clamp* or *zld*. We chose to measure the localization patterns of these three target genes because they showed significantly reduced expression in RNA-seq experiments. Localization of each early patterning gene was disrupted after depletion of maternal *clamp*.

We have added additional text regarding this phenotypic effect in the introduction, methods, and main text. The new imaging data are in Figures 1A-B.

Here a summary of our new results:

A) By comparing maternal depletion of CLAMP and ZLD, we found that the phenotypes caused by the maternal depletion of *clamp* are very similar to those caused by the maternal depletion of *zld,* including failed cellularization. Furthermore, embryos stall at the blastoderm stage when maternal *clamp* is depleted.

B) We characterized the expression pattern of the pair-rule genes *eve* and *run*, as well as the distribution of the cell adhesion glycoprotein Nrt because they showed significant expression reduction in RNA-seq data obtained after depleting maternal *clamp*. These results (Figure 1) also support the conclusion that the loss of maternal *clamp* in the early embryo is catastrophic because the seven segmentation stripes formed by the pair-rule genes fail to form.

How pure are the embryo collections? What is the proportion of embryos further than nuclear cycle 13,14 in the 0-2 hpf collections?

Thank you for the question about embryo collection which is a key issue. The embryo collection was precisely timed. To ensure that female flies laid all older embryos before we started our collections, we first starved flies for 2 hours in empty cages and discarded these first grape agar plates (Plate set #0).

When we collected embryos for the experiments, we put flies in the cages with grape agar plates including yeast paste to promote egg laying for 2 hours (Plate set #1). Next, at the two-hour time point, we replaced the first plates with a set of new plates (Plate set #2).

We kept Plate set #1 embryos (without any adult flies) to further develop for another 2 hours to obtain 2-4hr embryos. At the same time, we obtained newly laid 0-2hr embryos from Plate set #2. Therefore, this strategy successfully prevented the potential cross contamination of 0-2hrs embryos (Plate set #2) with the 2-4hr embryos (Plate set #1).

The description for embryo collections was added to Materials and methods.

“Embryo collections

To optimize egg collections, young (5-7 day old) females and males were mated.

To ensure mothers do not lay older embryos during collections, we first starved flies for 2 hours in the empty cages and discarded the first 2-hour grape agar plates with yeast paste (Plate set #0). When we collected eggs for the experiments, we put flies in the cages with grape agar plates (Plate set #1) with yeast paste for egg laying for 2 hours. Then, we replaced Plate set #1 with a new set of plates (Plate set #2) at the 2hr time point. We kept Plate set #1 embryos (without any adult flies) to further develop for another 2 hours to obtain 2-4hr embryos. At the same time, we obtained newly laid 02hr embryos from Plate set #2. Therefore, this strategy successfully prevented cross contamination between 0-2hr embryos (Plate set #2) and the 2-4hr embryos (Plate set #1).”

In addition, the validation of the knockdowns is buried in Figure 3 —figure supplement 1C,D, and is not convincing. This is a critical piece of information which is necessary to interpret the results for all remaining figures in the manuscript.

Thank you for bringing up this important concern. Both *clamp* and *zld* RNAi knockdown lines which we used were previously validated by multiple publications as described below. Furthermore, our qPCR results support efficient and specific knockdown of both factors. Unfortunately, the ZLD western blot we reported is the best one we obtained after multiple attempts. We have inquired with several laboratories but have been unable to obtain an aliquot of a ZLD antibody that works well on a western blot.

A) The *zld* RNAi line was made and published by the Rushlow lab (Sun et al., 2015; Yamada et al., 2019) and has been previously validated. See Methods in Sun et al., 2015: https://www.ncbi.nlm.nih.gov/pmc/articles/PMC4617966/#s3title

B) The *clamp* RNAi line has also been used previously and validated by Rieder et al. (2017). Below is the western blot from Figure 5A in Rieder et al. (2017) for the same stage embryos:

https://www.ncbi.nlm.nih.gov/pmc/articles/PMC5588930/figure/RIEDERGAD300855F5/

We added the following information regarding validation of fly lines to the Materials and methods section:

“Fly stocks and crosses

To deplete maternally deposited *clamp* or *zld* mRNA throughout oogenesis, we crossed a maternal triple driver (MTD-GAL4, Bloomington, #31777) line (Ni et al., 2011) with a Transgenic RNAi Project (TRiP) *clamp* RNAi line (Bloomington, #57008), a TRiP *zld* RNAi line (from C. Rushlow lab) or *egfp* RNAi line (Bloomington, #41552). *egfp* RNAi line was used as control in smFISH, Immunostaining and Imaging experiments. The MTD-GAL4 line alone was used as the control line in ATAC-seq and ChIP-seq experiments. Briefly, the MTD-GAL4 virgin females (5-7day age) were mated with TRiP UAS-RNAi males to obtain MTD-Gal4/UAS-RNAi line daughters. The MTD drives RNAi during oogenesis in these daughters: therefore, the targeted mRNA will be depleted in their eggs. Then MTD-Gal4/UAS-RNAi daughters were mated with males to produce embryos with depleted maternal *clamp* or *zld* mRNA targeted for ATAC-seq and ChIPseq experiments. The embryonic phenotypes of the maternal *zld*^-^ TRiP RNAi line were confirmed previously (Sun et al., 2015). Maternal *clamp*^-^ embryonic phenotypes of the TRiP *clamp* RNAi line were confirmed by immunofluorescent staining in our study (Figure 1A-B). Moreover, we validated CLAMP or ZLD protein knockdown in early embryos by western blotting using the Western Breeze kit (Invitrogen) and measured *clamp* and *zld* mRNA levels by qRT-PCR (Figure 1 —figure supplement 1B, C).”

2. The authors claim that CLAMP is a pioneer transcription factor that directly activates zygotically-transcribed genes (line 140), but in my opinion, this is not demonstrated in a convincing manner. First, the western blot presented in Figure 3 —figure supplement 1D doesn't allow for an evaluation of the presence of Zld at the protein level in CLAMP depleted embryos (see point 1 above).

Thank you for this comment. As addressed above, the ZLD Western blot is unfortunately not possible to redo due to lack of antibody availability from several labs.

Furthermore, the analysis presented in Figure 1 —figure supplement 1A, shows that there is a widespread change in expression for zygotically expressed genes upon CLAMP depletion irrespective of the level of CLAMP binding, suggesting that overall the observed changes correspond to indirect effects.

Thank you for bringing up the key issue of direct and indirect effects. Similar to the majority of transcription factors, CLAMP does not have a direct effect on transcription at all of its targets. Furthermore, in the absence of CLAMP, GAF can bind to some CLAMP target sites (Kaye, Cell Reports 2018; Rieder, 2017). Therefore, we would not expect a direct linear relationship between CLAMP occupancy and its impact on gene expression.

However, the density plot that in current Figure5-Supplementary Figure 1A does show that increased binding of CLAMP increases the chance that a gene will have its expression reduced in the absence of CLAMP.

Similar trends, but again not a perfect correlation, were observed in an analysis of Zelda occupancy and Zelda-dependent gene expression (Figure 2A-B in Harrison et al., 2011: https://journals.plos.org/plosgenetics/article?id=10.1371/journal.pgen.1002266).

As suggested by Review #3, we have replaced the density plot as a violin plot to improve the data visualization (current Figure5 B-C).

3. The manuscript doesn't contain critical information regarding how the ATAC-seq data were analysed. Did the authors use only short/long/all fragments for their analyses? This is important since one can then evaluate whether the authors are measuring open chromatin (short fragments) or nucleosome positioning (long fragments).

We apologize for not providing more detailed information describing the ATAC-seq analysis methods. We have now included a full description in Material and Methods and provided all ATAC-seq analysis code as Figure2-supplementary file 1.

“ATAC-seq analysis

Detailed ATAC-seq analysis pipeline code is provided as Figure2- supplementary file 1. Prior to sequencing, the Fragment Analyzer shows the library top peaks are in 180-190bp range, which is comparable to the embryo ATAC-seq online protocol (Haines, 2017). Demultiplexed reads were trimmed of adapters using TrimGalore

(Krueger, 2017) and mapped to the *Drosophila* genome dm6 version using Bowtie2 (v. 2.3.0) with option -X 2000. We used Picard tools (v. 2.9.2) and SAMtools (v.1.9, Li et al., 2009) to remove the reads that were unmapped, failed primary alignment, or duplicated (-F 1804), and retain properly paired reads (-f 2) with MAPQ >30. After quality trimming and mapping, the Picard tool reported the mean fragment sizes for all ATAC-seq mapped reads is between 125-161bp. As expected, we observed three peaks: (1) a sharp peak at <100 bp (open chromatin); (2) a peak at ~200bp (mono-nucleosome); (3) and other larger peaks (multi-nucleosomes). After mapping, we used Samtools to select a fragment size <= 100bp within the bam file to focus on open chromatin. Peak regions for open chromatin regions were called using MACS2 (version 2.1.1, Zhang et al., 2008). The peak number was summarized in Table 1 and peaks in each group were reported in Table 1 -Table supplementary 1. ENCODE blacklist was used to filter out problematic regions in dm6 (Amemiya et al., 2019). DiffBind with the DESeq2 method (v. 3.10, Stark and Brown, 2019) was used to identify differentially accessible regions.

The DA and non-DA number were summarized in Table 1 and DA peaks were reported in Table 1 -Table supplementary 2.

We used DeepTools (version 3.1.0, Ramírez et al., 2014) to generate enrichment heatmaps (CPM normalization) and average profiles were generate in DeepStats (Gautier RICHARD, 2020). Bigwig files generated by Deeptools bamCoverage (CPM normalization) is uploaded to IGV (Robinson et al., 2011) for genomic track visualizations (Figure 2B). Then, we used Homer (v 4.11, Givler and Lilienthal, 2005) for de novo motif searches (Figure 2D). Visualizations and statistical tests were conducted in R (R Core Team, 2014). Specifically, we annotated peaks to their genomic regions using R packages Chipseeker (Figure 2C, Yu et al., 2015) and we did gene ontology enrichment analysis using clusterProfiler (Yu et al., 2012). Boxplot and violin plot was generated using ggplot2 (Wickham, 2009) package. Intervene (Khan and Mathelier, 2017) was used for intersection and visualization of multiple peak region sets (Figure 2E).”

In addition, how are differentially accessible regions calculated? Is the genome binned? If so, how big are these bins? I am not sure that I fully understand how this analysis is done, and I don't know what "log concentration" refers to in Figure 2A, but I find difficult to reconcile the close to zero correlation values reported for the ATAC-seq datasets in the MTD and CLAMP depleted samples when the majority of dots in panel 2A are not different between the two conditions.

The differentially accessible regions were called using the DiffBind DEseq2 method, which is commonly used in ATAC-seq data analysis: https://rockefelleruniversity.github.io/RU_ATAC_Workshop.html

The genome is not binned in this method, but in the read counting step, DiffBind centers the peak at summit and extends 50bp upstream and downstream of the summit to count reads in a 100bp window.

The “log concentration” was automatically generated by DiffBind. Here is a detailed explanation from the software developer:

“For the X axis, "concentration" refers to the mean (normalized) number of reads across all the samples for that binding site. This is reported as a log2 value, so as you go from left to right, the overall binding affinity (read density) is doubling. “

The dark spot near the origin is a cluster of sites that have very low read counts and also do not change significantly (Figure 2A). The main dark region shows sites with increasing binding activity (high x-axis values) that do not change significantly between conditions (y-axis value close to 0). Both of the dense blue areas are shifted slightly below a fold change of 0 (y-axis), indicating a tendency to see more reads in the MTD group.

The pink points are "significantly differentially bound/accessible" sites. The absolute values of the fold changes are greater than 2 because the y-axis is also on a log2 scale indicating at least a 4-fold change in binding affinity. The red dots on the outer diagonal lines are sites that have no binding in one condition and substantial binding in the other condition.”

To make the plot easier to understand, we changed the x- and y-axis labels of Figure 2A to “MTD ATAC-seq (CPM)” and “Clamp-i ATAC-seq (CPM)”.

4. The genomic track plot presented in Figure 2B doesn't help in the interpretation of the data. First, the panel is missing the genomic coordinates, so one cannot determine what are the peaks that are presented in the figure. Second the panel shows multiple CLAMP binding sites that don't seem to present an open chromatin signature, suggesting that only a fraction of the binding is related to open chromatin in the control samples. It would be interesting, and, in my opinion, much more straightforward to test, how much of the binding that leads to open chromatin (irrespective of whether this would be scored as differential accessible regions, which might be confounded for the reasons outlined in points 1 and 2 above) occurs together, for example, with Zld binding.

We thank the reviewer for the suggestion, we have updated Figure 2B with all the information you suggest.

You are correct that all CLAMP binding sites are not involved in opening chromatin. For example, CLAMP binding peaks associated with introns are less likely to be involved in opening chromatin than those at promoters (Figure 2C). Moreover, we have integrated CLAMP and ZLD ChIP-seq and ATAC-seq peak types to define the relationship between CLAMP binding, ZLD binding and their ability to open chromatin (Figure 3A-B). Our major conclusion about the relationship between CLAMP and ZLD occupancy is that they often function together to open chromatin at promoters, where they regulate each other’s occupancy (Figure 2C, 3A-B and 4F).

5. I find the scatterplots and the regression analysis presented in Figure 2E very unconvincing. Although statistically significant, the correlations are extremely weak, and if the authors would compute R^2 this would be extremely low, highlighting that the majority of the variance observed in these measurements remains unaccounted for. So overall, in my opinion, most of the authors' conclusions in Figure 2 are not supported without further analysis.

We thank the reviewer for this point. We agree with literature which supports Reviewer #3 who states: “the ATAC-seq and RNA-seq are very different assays and as such it is to be expected that the fold change upon depletion of a factor should not be expected to be correlated in magnitude between assays, only the change in direction.” We therefore replaced the original scatter plot with a Venn diagram which shows the significant overlap between genes where CLAMP opens chromatin (ATAC-seq) and genes which are activated by CLAMP (mRNA-seq) (current Figure 2E).

6. I find the analytical strategy in Figure 3A-C inappropriate and difficult to understand. First, the authors use average plots that contain order of magnitude different number of regions across the comparisons. The manuscript does not contain specific details regarding how these plots are produced, but these can be severely affected by outliers, especially in the cases when the number of regions is low. Second, the author define four classes of regions, depending whether they are bound or not bound by the factor and differentially accessible, and a control region (not bound and not differentially accessible). The split in these classes is very confusing and difficult to follow throughout the rest of the manuscript. In my opinion, this might be easier to follow if the data were presented as heatmaps including the common set of regions then split into different classes.

We thank the reviewer for this comment. We chose to use average profiles because they have been used previously to study the role of Zelda in regulating chromatin accessibility. For example, the previously published work on Zelda regulating chromatin accessibility near its binding sites also divided peaks into 4 groups in the same way we reported (Figure 2). Schulz et al., 2015:

https://www.ncbi.nlm.nih.gov/pmc/articles/PMC4617967/

We apologize for not providing details of the analysis in the methods. As reviewers requested, we have now provided all of our code as a supplementary file.

Thank you for the suggestion to generate heatmaps instead of average profiles. We have now replaced most average profiles plots with heatmaps to allow the better analysis of the distribution of patterns at individual loci (Figure 3).

Of note, the authors use comparisons across the average plots to "validate" (line 231) or "confirm" (line 243) their analysis. I disagree that this is validation because the groups have been chosen presumably based on thresholding the same data, and a validation could only come from orthogonal data, which is not used here.

Thank you for pointing out that we should revise our word choice to be more accurate. We have revised the “validate” and “confirm” to “revealed” and “correspond.”

If the authors would want to use average plots, these should at least include a sharing area representing a confidence interval. Only one of the lines in Figure 6D shows such a shading, although that this represents is not explained. This significantly affects the majority of analyses and interpretations in Figures, 3,4,5,6.

Thank you for pointing this out. All average plots were generated in Deeptools and DeepStats used “--plotType se” to plot the average line with a confidence interval (see code for plot). However, when numbers of similar sites are large, it is not possible to see the shading because the confidence intervals are very small. The shading can be seen when there are fewer sites (e.g. Figure 6D: 30 sites). As we describe above, we have replaced most average profiles with heatmaps to provide more insight into the distribution of individual sites.

7. There is a significant difference in the shape of the average profiles for the ATAC-seq data for the fourth group in both comparisons in Figure 3A. This is meant to be the control group in both cases, of non-DA, non-bound peaks. Since the controls are qualitative different between the CLAMP and Zelda depleted experiments, I wonder whether strong conclusions can be obtained from the comparisons of the other average profiles. In this respect, the p-values in Table 1 need to be corrected for multiple testing. Under Bonferroni correction, the p-value of the overlap between DA Zld-bound and DA CLAMP-bound is not significant, which is in disagreement with the authors' conclusions (line 247-248). This also affects the motivation for the analysis in Figure 4 (lines 312-313).

Thank you for pointing out that the shape of the ATAC-seq data average profiles differs for among four groups in our previous analysis. Now, we have selected fragment sizes which are all less than 100bp, so we did not compare the shapes of these data. Moreover, we have replaced all average profiles in the previous manuscript to the heatmaps for better data interpretation.

We apologize for not being clear in describing the method used in the p-value calculation in Table 1 (now Table 4).

Briefly, the problem of gene overlap testing can be described by a hypergeometric distribution where one gene list A defines the number of white balls in a jar and the other gene list B defines the number of white balls that are drawn from a jar. Assume the total number of genes is N, the number of genes in list A is A and the number of genes in list B is B. The intersection between A and B is k.

Therefore, we calculated the probability of have k numbers in the intersection:

enrich_pvalue <- function(N, A, B, k)

{

m <- A + k

n <- B + k

i <- k:min(m,n)

as.numeric( sum(chooseZ(m,i)*chooseZ(N-m,n-i))/chooseZ(N,n) )

}

We added the code for the calculation in the supplementary ATAC-seq pipeline files.

8. Figure 3D is misleading since the scales in both plots are different and do not let the reader appreciate the extent of the changes for the different classes. It might help with the visualisation if the authors would use a violin plot, or if they would plot the data without the outliers and including notches for the boxes. In my opinion, the results in Figure 3D indicate that there is a downregulation of gene expression at CLAMP-bound genes irrespective of changes in chromatin accessibility. This would challenge the authors' own conclusions of a pioneer role for CLAMP at those sites that don't change accessibility. It is therefore unclear how the authors arrive at the conclusions in lines 294-296.

Thank you for pointing this out. We have now taken this figure out and moved all of the transcription data to the new Figure 5. Also, we plotted the relationship between CLAMP binding and transcription reduction after CLAMP depletion in a violin plot as you suggested. Statistical analysis of these plots supports our conclusions that there is a greater chance that a gene will be downregulated by CLAMP depletion if it is more strongly bound by CLAMP.

9. Does the list of motifs included in Figure 4A contain all the set of significant motifs found in these regions? Without this information, it is not possible to evaluate the statement in lines 319-320.

We thank the reviewer for this comment. Figure 4A only shows the top 3 most significant motifs found in these regions. The Homer software reported a long list of motifs and therefore we only showed the top three most significant motifs consistent with previous reports (Schulz et al., 2015). After generating heatmaps, we have performed new analysis combining motif calls and heatmaps where we include the top three most significant motifs for each cluster of sites (current Figure 3A).

Furthermore, this can be due to the fact that there is a significantly lower number of DA CLAMP-bound in 2-4 hours compared with the 0-2 hours data, which might affect the significance of these enrichments. In my opinion, a better alternative would be to show the number of Zelda events that occur in these regions. Without this information, the conclusion stated in this paragraph (lines 320-322) is not supported.

We thank the reviewer for this suggestion. To address this important concern, we have performed a new heatmap visualization of these data which shows both CLAMP and Zelda binding at clusters of sites where chromatin accessibility is regulated by each factor (Figure 3).

10. The authors state that the majority of Zelda binding sites are not affected upon CLAMP depletion (line 345). However, I find this statement puzzling, since Figure 4 —figure supplement 1A shows a significant effect in the level of Zelda binding upon CLAMP KD throughout most of the regions shown in the heatmap.

Thank you for this comment. However, this statement was based on the result of statistical analysis identifying significant differential binding of Zelda in the absence of CLAMP across replicates. The Diffbind DEseq2 method reported 274 (0-2hr) and 1,289 (2-4hr) statistically significant down-DB sites where Zelda binding decreased in the absence of CLAMP compared to MTD controls. Diffbind identified many more nonsignificant binding sites (0-2h: 3,144; 2-4h: 5,672). In this way, we determined that the majority of Zelda binding sites were not affected by CLAMP depletion in a statistically significant way.

To avoid confusion, we removed this specific statement from the text.

11. Related to the point above, it would be useful if the authors could clarify how the data presented in Figure 4 —figure supplement 1A and Figure 4 —figure supplement 1D are different. Also, please note that this figure is missing the labelling of the x-axis. This is also the case for Figure 4 —figure supplement 1E.

We thank the reviewer for this comment:

1) Figure 4 —figure supplement 1A shows all peaks ranked by their binding intensity compared with the input control.

2) Figure 4 —figure supplement 1D shows peaks plotted separately for upregulated, down-regulated, and non-significantly bound peaks/regions.

We apologize for missing information in the plot. All plots have been remade and the missing labels were added.

12. The conclusions reached by the authors in lines 354-356 regarding the effect of Zld in CLAMP binding are not supported since, as the authors acknowledge, the experimental design is confounded by the up-regulation of Zld in the 2-4 hours time point.

Thank you for this comment. Our conclusion regarding the effect of maternally deposited Zelda on CLAMP binding is supported at 0-2hr because 390 peaks were identified as differentially regulated by Diffbind DEseq2.

We acknowledge in the manuscript that the Zelda levels are recovering due to the expression of zygotic Zelda. However, our focus of this study is the maternal effects of both proteins.

We did remove this statement to avoid causing confusion.

13. I am unable to understand the interpretation of the results presented in Figure 4F and 4G. In any case, the results in Figure 4G might be confounded by the increased in Zld expression at 2-4 hours, as mentioned by the authors before.

We apologize for the data complexity in combining ATAC-seq and ChIP-seq analysis.

We have updated the analysis by generating a heatmap as you have suggested (Figure 3). The heatmap has greatly increased the ability to interpret different classes of ATACseq and ChIP-seq binding sites and supported our conclusions.

14. Figure 5A lacks genomic coordinates, which makes it impossible to interpret the plot. In addition, the scales of the signal are also not readable, which makes it impossible to evaluate the robustness of the binding represented and the comparison.

We apologize for missing the information on the figure. We have added the genomic coordinates to the plot and remade the figure in illustrator for improve readability of the labels.

Related to this figure, does the difference in peak size depend on the number of individual binding events?

Thank you for this question. The difference in peak size could be associated with the number of individual binding sites since the broader peak regions contains multiple motifs. Furthermore, we have previously reported that broader CLAMP ChIP-seq peaks correlate with the presence of multiple clustered motifs (Soruco et al., 2013).

I am unable to follow the results presented in Figure 5 —figure supplement 1E,F, and the interpretation of the results of Figure 5E.

Thank you for this statement.

1) Figure 5 and Figure S1E and F: These plots show the number of protein binding motifs in Down-DB and Non-DB sites when these sites are defined based on the depletion of the other protein.

A) We found that Zelda motifs are more significantly enriched at sites where CLAMP occupancy is decreased after depleting Zelda.

B) Similarly, CLAMP motifs are significantly enriched at sites where CLAMP is required for Zelda binding.

Therefore, we conclude that CLAMP and Zelda regulate each other’s binding via directly binding to their own binding sites because the motifs for the required protein are enriched. This supports the conclusion that CLAMP and Zelda directly regulate each other’s binding. To avoid confusion, we removed this part of our analysis in the results.

2) Figure 5E (current Figure 5D): We apologize for the data complexity which results from combining ChIP-seq and RNA-seq analysis but it is important to integrate these two data types. Here, we plot transcription data for genes where TF occupancy is altered by the reduction of the other TF.

Briefly, the y-axis shows CLAMP target gene expression reduction after depleting Zelda (log 2 fold change compared to control). The x-axis shows differentially bound and non-differentially bound genes for 0-2hr (left) and 2-4hr (right) embryos. We found a significant reduction of CLAMP target gene expression reduction after Zelda depletion only in the differentially bound (down-DB) group at the 0-2hr time point. Therefore, genes where Zelda impacts CLAMP binding show altered target gene expression.

15. I find it difficult to understand the statement in lines 458-459 because I do not understand what is the nature of the interdependent relationship between Zld and CLAMP binding to chromatin.

Thank you for this statement. As suggested by Review #3, we changed

“interdependent” to “cooperative” throughout the manuscript. In our model, CLAMP and Zld help each other to binding to promoters of target genes which regulates target gene expression. Many of these target genes encode DNA binding transcription factors which drive genome activation.

16. It is unclear whether the data in Figure 6C refers to the dependent or independent sites, since both seem to gain accessibility upon Zld depletion. I find this observation difficult to reconcile with the results presented in Figure 4 —figure supplement 1A,E that suggest that Zld depletion leads to an overall reduction of CLAMP binding. I would have expected then to observe a loss of accessibility, but not a gain. How do the authors explain this puzzling observation?

Thank you for bringing up this important point. One possible explanation could be technical and related to the ATAC-seq technology: the increased accessibility of these regions upon Zelda depletion could be because Zelda is simply stably bound at these sites which inhibits access of the Tn5 enzyme used in ATAC-seq. Removing Zelda makes these sites more accessible for transposition. We have added this statement to the Results section:

“Interestingly, the accessibility slightly increases upon the loss of ZLD in CLAMP downDB at 0-2hr (Figures 5E). An active TF binding to DNA could prevent Tn5 cleavage at those regions (Yan et al., 2020). Therefore, loss of ZLD and CLAMP binding could result in a perceived accessibility increase, as measured by ATAC-seq.”

In any case, since the gain in accessibility seems to be independent of CLAMP binding, since it occurs in both groups, can the authors be confident that this is biological and not due to technical differences between the libraries? What is the overlap between the sites that are reported here and those reported gaining accessibility in Schulz et al.?

Thank you for this question. Please note that we did not generate libraries for the

ATAC-seq data from zld-i embryos. These data were obtained from Hannon et al., (2017). In their processed data, there are sites which show significant differential accessibility in both directions upon Zelda germline depletion.

17. The results in Figure 6D are also very difficult to interpret, especially given the limited effect of Zld depletion on CLAMP binding at 2-4 hours. How do the authors explain these results?

Thank you for this question. In Figure 6D, we focused on peaks that show differential binding. Although the effect of ZLD depletion on CLAMP binding is modest, we can still capture the profile for the most significant peaks (n=30). Chromatin accessibility was decreased by depleting Zelda at regions where CLAMP occupancy is also significantly reduced. These data support a model in which Zelda increases chromatin accessibility to promote CLAMP recruitment.

18. I don't understand the authors reasoning for the statement in lines 490-492. The authors' own analysis shows that the overlap between the set of downregulated genes at 2-4 hours is not better than one could expect by random chance (Figure 6F). How do the authors then conclude that there is co-regulation for hundreds of genes after ZGA?

Thank you for this question. The *p*-value does not show significance of the overlap between downregulated genes at 2-4hr compared to 0-2hr. However, we did not want to ignore that there are 373 genes which overlap by concluding that Zelda and CLAMP become independent of each other after ZGA. We removed this plot and replace it with the current Figure 5 which includes all of the transcription-related analysis in the new version of the manuscript.

Reviewer #2:In the current manuscript under review, Duan et al. address the question of the role of GA-repeat binding factor CLAMP on the process of ZGA. The question of ZGA, particularly that of which pioneer factors establish patterns of chromatin accessibility and promote the expression of the first zygotic transcripts has received heavy attention in recent years. Notably, although another pioneer, Zelda, has a critical role for driving ZGA for a subset of zygotic genes by several measures, the vast majority of genomic locations either require a combination of Zelda and another factor, or another factor entirely. Several prior studies have pointed to enrichment for a GA-repeat motif within this class of sites. Identifying and characterizing the role of such a second maternal pioneer would represent a significant advance for the field as well as more broadly across biological fields as the question of pioneering touches on several key aspects of transcriptional regulation and epigenetics.While Duan et al. present data that (1) CLAMP binds its motif even in the nucleosome-associated state; (2) CLAMP loss of function leads to some amount of reduced chromatin accessibility; (3) Some CLAMP and Zld DNA binding is interdependent; (4) Loss of CLAMP function affects gene transcription-- the manuscript in its current state is far from suitable for publication. My primary concern is that the data presentation of the genomics studies is extremely difficult to follow, that supporting data tables are either incompletely annotated or missing, in many cases it is nearly impossible to read the plot labels in the figures, and that the biological significance of the observations is not fully substantiated. In addition, certain controls have not been provided or even incorporated into experimental design. Also, there are issues with the presentation of the study, with factually incorrect statements and missing or unclear description of methods, and missing references (in some cases leading to factually incorrect statements).This could be an important paper and it is therefore important that the presentation is watertight. I provide the comments below fully aware of current constraints on daily life, and in the spirit of wanting to minimize additional work for the authors. I think that overall the data already exists to improve the manuscript (or at least it should). But there is a fundamental question of whether the data are over-interpreted, and whether the effect of Clamp is as significant as the authors claim, at least within the framework of the process of ZGA.

We thank the reviewer for their careful assessment which has helped us greatly to significantly improve the manuscript. We added supporting data tables and missing information, changed figure labels, and provided additional experiments and computational approaches to strengthen the manuscript. All of your questions are addressed below:

1) The presentation of the genomics data analysis is very difficult to follow. I inspected the bigWig files for the ATAC data and had a hard time finding genomic regions where there is clear-cut evidence for CLAMP's role as a pioneer factor. Loading up the four ATAC conditions (two timepoints each control or clamp-i), as well as the Rieder CLAMP ChIP (NC14) and the 3h Harrison Zld ChIP-seq, I can find only a handful of regions where CLAMP has a clear all-or-nothing effect on chromatin accessibility, and these (few) sites are at regions where there is little Zld binding. These sites I did find by scrolling through nearly the entire genome are: 3' to CG11448, within iab-8, and possibly at the promoters of Vsx1 and 2. There are, however, numerous examples of regions where a 'differential enrichment' analysis could possibly yield a statistically significant difference between control and clamp-i, but there remains substantial accessible chromatin in the knockdown conditions. This latter phenomenon cannot be construed as evidence for pioneer activity, since it is expected that in the absence of the pioneer, the locus would be inaccessible. I am left with the question of whether the effect of Clamp on chromatin accessibility is oversold in this study.

We thank the reviewer for careful assessment which we have addressed as follows:

We re-sequenced our ATAC-seq data to obtain more reads and therefore higher quality data. We have added details of data analysis in Materials and methods and the code in the supplementary file. Here are some key points of the revision:

1) We have updated the candidate region in the current Figure 2B to more accurately show a better represented region with differential accessibility based on computational analysis.

2) We added an average profile showing the significant reduction of ATAC-seq reads in clamp-i embryos at DA sites compared with controls (Figure 2A).

3) We also added a heatmap of different classes of ATAC-seq peaks (Figure 3) which includes binding profiles for both CLAMP and Zelda. Furthermore, CLAMP binding is stronger at DA sites bound by CLAMP compared to nonDA sites bound by CLAMP.

4) We have added Tables to the manuscript and summarized numbers and fractions of peaks displaying an effect to improve the data presentation. Moreover, in supplementary tables, we reported peaks locations of DA called by DiffBind, and all 4 defined CLAMP or ZLD-related classes.

Overall, we observe a significant reduction of read enrichment at differentially accessible regions upon the *clamp* depletion. Similar to Zelda, CLAMP has both direct and indirect effects on chromatin accessibility and these effects do not occur at all factor binding sites as we state in the manuscript. For example, CLAMP has stronger effects on chromatin accessibility at promoters than at introns (Figure 2C).

The example regions plotted in the Figures also reveal potential issues in the analysis or interpretation of data: Figure 2B, CG11023: I had questions about what was going on in this plot which were cleared up by checking the bigWig files. For instance, I was curious why the light blue peak region indicator included regions with no ATAC signal in either control or clamp-i. Why also is there Clamp ChIP signal in this region? Upon inspection of the data, this plot shows base one of chr2L, and the blank region in the ATAC is presumably due (understandably) to mapping issues at the very telomeric end of the chromosome. Why is a peak called here? Why does the peak end within a peak of ATAC signal and not include this whole region? Significantly more concerning is that when I examine this region, my conclusion is that this whole region is likely very low signal that I would be reluctant to score both as "open" as well as "bound by Clamp". On the basis of this, I am reluctant to say that the bioinformatic analysis has been performed with sufficient rigor.

We apologize for not carefully choosing a representative region. We updated the region (promoter of *Mod(mdg4)*) in the current Figure 2B. We also made sure this example region is a differentially accessible region and a region where CLAMP is differentially bound to demonstrate the direct impact of CLAMP binding on chromatin accessibility. Moreover, we also found that ZLD is also differentially bind to this region upon CLAMP depletion (data is not shown in IGV plot). Moreover, we have now added a section of ATAC-seq and ChIP-seq data integration to our Methods and Materials section and provide all peaks and peaks in each sub-group in bed format as supplementary tables.

“ATAC-seq and ChIP-seq data integration

We first selected the top 6,000 ATAC-seq peaks based on the count per million value rank from our ATAC-seq data or the ATAC-seq in Hannon et al. (2017). Then, we used Bedtools (Quinlan and Hall, 2010) intersection tool to intersect peaks in CLAMP ChIP-seq binding regions with CLAMP DA or non-DA peaks. Based on the intersection of the peaks, we defined 4 types of CLAMP related peaks: (1) DA with CLAMP, (2) DA without CLAMP, (3) Non-DA with CLAMP, (4) Non-DA without CLAMP (Figure 3A, Figure 3—figure supplement 1A, and Table 2).

Similarly, we defined ZLD related peaks by intersecting ZLD DA or non-DA peaks and ATAC-seq datasets (Hannon et al., 2017; Soluri et al., 2020) from wildtype (wt) and *zld* germline clone (*zld-*) embryos at the NC14 +12 min stage. Specifically, we defined four classes of genomic loci for ZLD-related classes: (1) DA with ZLD, (2) DA without ZLD, (3) Non-DA with ZLD, (4) Non-DA, without ZLD (Figure 3B and Table 2). We used DeepTools (version 3.1.0, Ramírez et al., 2014) to generate enrichment heatmaps (CPM normalization) and average profiles for each subclass of peaks (Figure 3A-B). We used Homer (v 4.11, Givler and Lilienthal, 2005) for de novo motif searches. Peaks locations in each CLAMP or ZLD-related category were summarized in Table2 – Source Data 1.”

Admittedly, this is based on one example image, but I would also point out that the authors have both only provided limited example regions, and have not provided a sufficiently documented 'peaks list' that includes regions that they feel are (1) bound by CLAMP, (2) bound by Zelda, (3) score as a member of the various groupings used to compare regions throughout the text (e.g. DA-Clamp bound, et cetera). The peaks list that the authors do provide is in a strange format and the column labels are not included in that file (nor can I find anywhere a description of that file, but I may have missed that in the submission materials). Nevertheless, it does not appear to indicate membership in any of the different classes from what I can tell.

Thank you for the suggestion. During the revision, we have re-sequenced 3 ATAC-seq libraries that had lower than 5M usable (uniquely aligned) reads in our original analysis. Now after re-sequencing, we reached on average 25M reads per ATAC-seq sample (Figure 2-supplemental table 1). We have re-analyzed the data and updated all the plots in the manuscript. We have added Figure 2-supplimentary file1 and Figure 4supplimentary file1 to show the pipeline we used for ATAC-seq and ChIP-seq analysis. The code in each step and plots for each figure are all included. Moreover, we summarized peak number and fraction in main Table 1-4 and provided all peaks and peaks in each sub-class in bed format as supplementary tables.

It is similarly difficult to evaluate the conclusion that CLAMP has anything at all to do with ZGA (see below). Specifically, however, to the bioinformatics analysis: when RNAseq data is analyzed, is it limited to zygotic genes only (as defined either in DeRenzis 2007, or in the Li paper cited in the manuscript?), and is the magnitude of the effect large enough to warrant the conclusion that Clamp is required for ZGA?

Thank you for this question. We only separated maternal and zygotic genes when visualizing the impact of CLAMP impact on each category. We used all genes to overlap with CLAMP and ZLD DB, nonDB and to their bindings. Also, we have now moved all of the transcription data to the new Figure 5. For example, Figure 5B-C show that genes strongly bound by CLAMP showed a significant (*p* < 0.001, Mann-Whitney U-test) level of gene expression reduction after *clamp* RNAi than weakly bound or unbound genes. We also observed similar results (*p* < 0.001, Mann-Whitney U-test) on the ZLD strong binding genes with expression reduction upon *zld*- in germline clone embryo RNA-seq (Combs and Eisen, 2017). We noted the gene number in each binding group (strong, weak and none binding) on the plot and it shows similar numbers between CLAMP (eg. strong = 250, 463 in 2 time points) and ZLD (eg. strong = 207, 436 in 2 time points) regulated genes. Therefore, we showed the direct binding of CLAMP or ZLD to genes regulates their transcriptional activation in a similar magnitude during ZGA.

Moreover, we plotted the relationship between CLAMP binding and transcription reduction after CLAMP depletion in a violin plot as you suggested. Statistical analysis of these plots supports our conclusions that there is a greater chance that a gene will be downregulated by CLAMP depletion if it is more strongly bound by CLAMP.

For comparison, loss of Zelda function results in near zero transcripts produced from a subset of zygotic genes (and corresponding elimination full stop of chromatin accessibility at those loci). I'm worried that the authors are placing too much weight on "significant" p-values without considering if the magnitude of the effect supports the stated conclusions. If the effect of clamp-i is minimal on transcription and chromatin accessibility, which it may be based on my limited examination of the raw data, I see no way to justify the conclusion that Clamp has any major role in ZGA.

Thank you for the question. We have compared the impact of ZLD binding on gene expression and CLAMP binding on gene expression in the current Figure 5 B-C and they show very similar impact.

Moreover, we have performed new imaging experiments on early patterning genes which show similar phenotypes for embryos depleted of either CLAMP or ZLD (Figure 1). Like Zelda-depleted embryos, CLAMP-depleted embryos also show failed cellularization consistent with a defect in ZGA. Interestingly, CLAMP has a stronger role in promoting Zelda occupancy than Zelda does in promoting CLAMP occupancy (Figure 4A). Therefore, we are convinced that CLAMP promotes ZGA in early embryos.

I also have a difficult time finding any Zld-bound loci that convincingly show loss of accessibility in the clamp-i data.

Thank you for this comment. We have now re-sequenced our ATAC-seq data which has improved their quality. In the new Figure 3A we show that CLAMP regulates the accessibility of sites that are bound by Zelda and have Zelda motifs.

Reviewer #3:The manuscript submitted by Duan and Rieder et al. describes, for the first time, how the CLAMP transcription factor acts as a pioneer TF in the fly embryo. They demonstrate that CLAMP can bind to nucleosome-bound DNA and that it binds to and generates accessible chromatin at a set of gene promoters in the early embryo, and that without this activity these genes fail to be transcribed during ZGA. They further describe fascinating cooperativity between CLAMP and ZLD, a previously identified pioneer TF in the fly embryo. Their results are compelling and rigorous, and the work will be of broad interest to both the developmental biology and transcription biology fields.

We thank the reviewer for their careful assessment to help us to improve the manuscript. We have performed new experiments, re-sequenced our ATAC-seq libraries, added supporting data tables, changed figure labels, and provided more experimental/bioinformatics details to strengthen the manuscript. All your questions are addressed below.

One concern is in Figure 2E, This scatterplot and correlation is not particularly convincing. The fact that the positive correlation is very minor needs to be emphasized properly in the text. Moreover, in our opinion, there is a better way of doing this. ATAC-seq and RNA-seq are very different assays and as such it is to be expected that the fold change upon depletion of a factor should not be expected to be correlated in magnitude between assays, only the change in direction. The dynamic range of change you can expect in RNA-seq is much greater than that in ATAC-seq because mRNA is much more abundant than its cognate DNA for transcribed genes. We think the authors should simply display a Venn diagram of genes/promoters that move in the same direction, i.e. what fraction of the genes have the same directionality of change. We do not think that comparing the magnitude of these changes is particularly useful or informative in this case.

Thank you for your constructive suggestion. We have followed your advice and replaced the correlation plot with a Venn diagram to show overlap between ATAC-seq and mRNA-seq data at the gene level.

[Editors’ note: what follows is the authors’ response to the second round of review.]

Reviewer #3 (Recommendations for the authors):In the revised manuscript, Duan and colleagues have addressed some of the issues that were raised upon the original review. The authors have generally improved the presentation of their data and have rendered the results easier to interpret. However, despite these improvements, upon inspection of the differential enrichment analysis, the magnitude of effect of Clamp on differential chromatin accessibility is significantly overstated.It is appreciated that the authors re-sequenced some of their lower depth samples for the resubmitted version. It is also appreciated that the authors have now provided the annotated tables from the differential enrichment analysis. In my original review, I mentioned that I manually searched nearly the entire genome while struggling to find more than a few examples of convincing loss of ATAC signal in the clamp-i data. I have now reviewed the differential enrichment analysis ("Table 4-source data 2", referred to as "Table 1 Source Data 2" in the text (line 681)) and note the following issue:- The authors appear to have relied not on the FDR but rather on individual, independently calculated p-values for reckoning the number of differentially accessible peaks in the differential enrichment analysis. Table 1 reports 76 "Up", 1675 "Down", and 9465 "None" effects on accessibility in clamp-i embryos versus control. In the supplied source data, it is clear that the authors set a p-value cutoff of 0.05 for calculating these numbers. What isn't mentioned in the text is that this cutoff corresponds to a 32% FDR. Typically, a 5% FDR rate is chosen to minimize incorrect rejections of the null hypothesis that arise due to multiple testing. Using this standard, the total number of differential peaks in the 0-2 hour comparison is only 95, with 73 sites showing a reduction, and 22 showing an increase. Again, a manual inspection of a sampling of these regions shows marginal differences in magnitude between the few regions that do pass significance testing at 5% FDR.Even fewer regions are differentially accessible in the 2-4 hour sample at a 5% FDR (total = 54, 33 "down", 21 "up").On the basis of this observation, it would be hard to argue that CLAMP is playing a major role in regulating chromatin accessibility at ZGA. To me, this substantially casts doubt on the central premise of this manuscript and in fact suggests that CLAMP has only a minor effect on accessibility at this time.

We appreciate the reviewer’s comments and concerns. Because embryos were pooled for ATAC-seq within a 2-hour time interval to match our ChIP-seq data, we do see variability among sample replicates. Therefore, fewer peaks were identified after multiple hypothesis correction because many peaks are only altered in one replicate.

Therefore, consistent with ENCODE guidelines, we reanalyzed our ATAC-seq data using a more stringent FDR cutoff. We updated several parameters for each tool:

Mapping

bowtie2 \

--local \

-p16 -t --very-sensitive-local \

--phred33 -N 1 -X 2000 \

Current, changed the local alignment:

bowtie2 \

-p16 -t --very-sensitive --no-mixed --no-discordant \

--dovetail -X 2000 -k 2

Peak calling MACS2

macs2 callpeak -f BAMPE -g dm --keep-dup all -nolambda -q 0.01 --cutoff-analysis

Current:

macs2 callpeak -f BAMPE --gsize dm --qvalue 0.01 --call-summits

Moreover, we found that multiple publications used FDR <0.1 for ATAC-seq data analysis including previous analysis of CLAMP in cell lines ( Samata et al. *Cell* 2020, MOF;, Albig et al *NAR* 2018CLAMP in S2/Kc cells). Therefore, we updated our DiffBind analysis using a newer version (v.3.12) that normalizes sample library size by “dba.normalize” and defined our differential peaks with FDR<0.1.

Using this cutoff, we obtained 277(0-2 hours) and 50 (2-4 hour) loci that are differentially accessible in our ATAC-seq data. We further restricted to loci that are bound by both CLAMP and ZLD for the downstream analysis (Figure 4). Please see Figure2-supplementary table 1 and Table 2-Source Data1 for peak location details.

Because our number of CLAMP-dependent differentially accessible peaks is reduced in our new analysis, we have performed an intensive revision to remove all statements that might sound overstated and discussed the clear difference between CLAMP and ZLD regarding the role they play in chromatin accessibility. We have clarified that ZLD functions at many more sites than CLAMP and have not defined CLAMP as a pioneer factor but as a factor with “pioneer-like” function. However, CLAMP function promotes ZLD recruitment is critical for zygotic genome activation. Moreover, many CLAMP target genes are key early TFs and components of the *Hox* cluster which likely explain the early patterning defects and lethality we report (Figure 1A).

CLAMP also regulates the chromatin accessibility of the *zld* gene locus itself.

Here is an example from the manuscript:

“Although we have demonstrated an instrumental role for CLAMP in defining a subset of the open chromatin landscape in early embryos, our data show that CLAMP does not increase chromatin accessibility at promoters of all zygotic genes independent of ZLD. Consistent with our results in the early embryo, CLAMP was found to regulate chromatin accessibility at only a few hundred genomic loci in male S2 (258 sites) and female Kc (102 sites) cell lines (Albig et al., 2019). Unlike ZLD which plays a global role in regulating chromatin accessibility at promoters throughout the genome, depletion of CLAMP alone mainly drives changes at promoters of specific genes that often encode transcription factors which are important for early development, consistent with phenotypic data.”

[Editors' note: further revisions were suggested prior to acceptance, as described below.]

The authors have made a concerted effort to address the reviewer's concerns, and save for the remaining minor issues below, the manuscript is suitable for publication. While the reanalysis of the data has led to the conclusion that Clamp does not alter chromatin accessibility at as many sites in as non-redundant a way as Zelda, the work does document an interesting and critical interplay of pioneer transcription factors in early embryonic development, and it begins to understand the molecular underpinnings of that interplay. We think this work will be of broad interest and will help clarify how transcription factors act to establish chromatin accessibility and set-up the first steps in early embryonic transcription regulation.

We thank all of the reviewers and editors for your constructive comments and suggestions which have helped us greatly improve our manuscript. We have addressed the remaining minor issues and edited the manuscript language thoroughly for clarity.

1) Clamp as Pioneer: the authors have convincingly shown that Clamp binds to nucleosomal DNA using gel shift assays, and this result alone is probably sufficient to call it a pioneer factor in our view. However, the authors have also convincingly shown that the scope of Clamp pioneering accessibility of chromatin is very small compared to Zelda, but that like Zelda, loss of function is catastrophic in terms of overall development. Any use of "pioneer-like" can be replaced with "pioneer'. We also recommend that the authors carefully edit the Discussion to accurately describe the magnitude of Clamp's effect on accessibility, and to update the summation of results pending the outcome of points 2 and 3 below.

Thank you for your support of our findings. We replaced the “pioneer-like” with “pioneer” throughout the manuscript and edited the discussion about the role of CLAMP in modulating chromatin accessibility and emphasized the more targeted yet essential function in zygotic development:

“Although we have demonstrated an instrumental role for CLAMP in defining a subset of the open chromatin landscape in early embryos, our data show that CLAMP does not increase chromatin accessibility at promoters of all zygotic genes independent of ZLD. Consistent with our results in the early embryo, CLAMP regulates chromatin accessibility at only a few hundred genomic loci in male S2 (258 sites) and female Kc (102 sites) cell lines. Unlike ZLD, which plays a global role in regulating chromatin accessibility at promoters throughout the genome, depletion of CLAMP alone mainly drives changes at promoters of specific genes that often encode transcription factors that are important for early development, consistent with phenotypic data. These findings indicate that CLAMP and ZLD regulate ZGA in different ways: ZLD mediates chromatin opening globally, while the CLAMP functions in a more targeted way at certain essential early TF genes. However, both proteins are critical to ZGA and loss of either is catastrophic in terms of overall embryonic development.”

2) The reviewers agree that part of the new analysis presented in Figure 3 was not performed in an ideal manner to support the conclusions. The observation at line 245, for instance, is premature:"Depletion of either maternal zld or clamp mRNA altered the genomic distribution of CLAMP and Zld: both factors shifted their occupancy from promoters to introns."We request the authors either repeat this analysis more rigorously or eliminate the section entirely. The current analysis is performed by comparing independently called peak lists and placing emphasis on regions that are present or absent in each set. This approach is highly susceptible to thresholding artifacts associated with peak calling. All reviewers agree that a more rigorous approach would be either to perform this analysis on a single, union peak set followed by differential enrichment analysis, or coverage data between different treatments could be compared directly by generating XY-scatter plots of summed reads in each peak from a union peak list. If the conclusion of this section is correct, the genomic regions of interest should be significantly off the diagonal, and this can be statistically addressed.

We thank reviewers for your comments on analysis methods related to Figure 3C-D. We used an approach from a widely-cited R package ChIPseeker (Yu et al., 2015) to compare peak set genomic distribution in multiple samples, but now we realized the conclusion we made from this single analysis is premature based on your comments.

We agree that Figure 3C-D can only indicate that the peak distribution is different in each genotype but cannot statistically determine a significant shift in occupancy from promoters to introns.

Thank you for your suggested analysis approach. Instead of generating a single union peak list for differential analysis as reviewers suggested, our analysis used the DiffBind package (Stark R and Brown G, 2011) that overlaps and merges peak sets across compared datasets, counting reads in overlapped intervals and uses DESeq2 to identify statistically significantly differentially bound sites. Therefore, we directly performed a new analysis for the genomic distribution of up and down-regulated differentially bound (DB) peaks identified using DiffBind/DEseq2.

We have added these new analysis results for all up/down-DBs in Figure 3-supplementary Figures 3 to replace Figure 3C-D. We removed text in the previous section and moved the text after the introduction of the differential binding analysis:

“Moreover, depletion of either maternal *zld* or *clamp* mRNA altered the genomic distribution of CLAMP and ZLD: the most common pattern we observed was that promoter-bound peaks were lost (down-DB) and peaks in introns were gained (up-DB) (Figure 3-figure supplement 3).”

3) The authors demonstrate that the knockdown efficiency of Zld RNAi is poor during the 2-4h timepoint (e.g. Figure 3, Fig. Supp. 1B). We caution the authors from drawing any strong conclusions about the effect of Zld on Clamp in the 2-4h time period. Please consider revising or eliminating the text beginning at line 262, where the weak effect of Zld on Clamp binding at 2-4 hours can possibly be attributed to incomplete knockdown.

Thank you for the suggestion. This section is now revised to eliminate any potential overstatement:

“390 (0-2 hr) and 30 (2-4 hr) CLAMP down-DB sites were found upon loss of ZLD (Figures 3D, Figure 3-figure supplement 1E, and Table 1). We identified very few sites where CLAMP occupancy increases after *zld* RNAi (up-DB sites: 0-2hr: 54, 2-4hr: 3).”

4) For most of the heatmaps throughout the manuscript: the titles of the heatmaps incorrectly refer to "peaks", regardless of the data type presented in the heatmap. This can be confusing since the y-axis of the heatmap is some set of "peaks," and the data presented in the heatmap is ATAC-seq coverage or ChIP-seq coverage for a particular factor/genotype/timepoint. To improve readability, please revise heatmap plots to indicate the peak set on the y-axis, and relevant sample information in the header/title of each plot.

Thank you for the careful assessment. We have now updated all heatmap titles and axes to reflect the data types.

For example, we replaced “CLAMP peaks” with “CLAMP occupancy” on the title and “ChIP-seq signal intensity” on the y-axis/color key.

We revised “ATAC-seq peaks” with “Chromatin accessibility” in the title and “ATAC-seq signal intensity” on the y-axis/color key.

5) Paragraph beginning at line 341. Here, the authors are examining "gene expression changes caused by depleting maternal Zld at genes where CLAMP regulates Zld binding." The next sentence, however, talks about "genes where Zld regulates CLAMP binding." (Genes where "CLAMP regulates Zld binding" are never mentioned again.) This makes the logic of this paragraph difficult to interpret. Please revise.

Thank you for pointing this out. This paragraph has been revised in the manuscript to the following:

“To investigate whether CLAMP and ZLD could regulate each other’s binding to precisely drive the transcription of target genes, we plotted the gene expression changes caused by depleting maternal *zld* or *clamp* at the genes closest to where they regulate each other’s binding (Figure 4E & Figure 4F). The depletion of maternal *zld* significantly (*p* = 4.3e-5, Mann-Whitney U-test) reduces the expression of genes where ZLD regulates CLAMP binding (down-DB) more than sites where CLAMP binds independently of ZLD (non-DB) (Figure 4E). Therefore, ZLD may specifically regulate zygotic genes at which ZLD promotes CLAMP binding. Also, compared to genes where ZLD binds independent of CLAMP, genes where ZLD binding is regulated by CLAMP had a significant (*p* < 0.001, Mann-Whitney U-test) expression reduction after *clamp* RNAi at both 0-2hr and 2-4hr time points (Figure 4F). Thus, CLAMP may regulate the transcription of genes targeted by ZLD by promoting ZLD binding.”

6) In general, the reader has to work hard to clearly interpret the results section of this manuscript, particularly for Figures 4-5. Please consider editing the text related to Figures 4-5 for clarity.

Thank you. We have edited the Figures 4-5 text (highlighted in manuscript) for clarity and accessibility.